# Leptin receptor-expressing neuron Sh2b1 supports sympathetic nervous system and protects against obesity and metabolic disease

Lin Jiang[1], Haoran Su[1], Xiaoyin Wu[2], Hong Shen [1], Min-Hyun Kim[1], Yuan Li[1], Martin G. Myers Jr [1,3], Chung Owyang[1,2] & Liangyou Rui [1,2✉]

Leptin stimulates the sympathetic nervous system (SNS), energy expenditure, and weight loss; however, the underlying molecular mechanism remains elusive. Here, we uncover Sh2b1 in leptin receptor (LepR) neurons as a critical component of a SNS/brown adipose tissue (BAT)/thermogenesis axis. LepR neuron-specific deletion of *Sh2b1* abrogates leptin-stimulated sympathetic nerve activation and impairs BAT thermogenic programs, leading to reduced core body temperature and cold intolerance. The adipose SNS degenerates progressively in mutant mice after 8 weeks of age. Adult-onset ablation of Sh2b1 in the mediobasal hypothalamus also impairs the SNS/BAT/thermogenesis axis; conversely, hypothalamic overexpression of human SH2B1 has the opposite effects. Mice with either LepR neuron-specific or adult-onset, hypothalamus-specific ablation of Sh2b1 develop obesity, insulin resistance, and liver steatosis. In contrast, hypothalamic overexpression of SH2B1 protects against high fat diet-induced obesity and metabolic syndromes. Our results unravel an unrecognized LepR neuron Sh2b1/SNS/BAT/thermogenesis axis that combats obesity and metabolic disease.

[1] Department of Molecular & Integrative Physiology, University of Michigan Medical School, Ann Arbor, MI 48109, USA. [2] Division of Gastroenterology and Hepatology, Department of Internal Medicine, University of Michigan Medical School, Ann Arbor, MI 48109, USA. [3] Division of Metabolism and Endocrinology, Department of Internal Medicine, University of Michigan Medical School, Ann Arbor, MI 48109, USA. ✉email: ruily@umich.edu

Adipose hormone leptin critically regulates body weight and metabolism, and disruption of leptin/leptin receptor (LepR) signaling results in morbid obesity and severe metabolic disease[1]. LepR is widely expressed in the hypothalamus, including the preoptic area (POA), lateral hypothalamus, dorsomedial hypothalamus (DMH), ventromedial hypothalamus, and arcuate nucleus (ARC)[2,3]. Leptin exerts its anti-obesity action by activating LepR signaling in hypothalamic energy balance circuits[4], but it remains elusive whether leptin regulates energy expenditure vs energy intake by similar or discrete pathways. Leptin signaling is mediated by tyrosine kinase JAK2 that interacts with long-form LepRb[1]. Of note, a number of negative regulators of JAK2, including SOCS3, PTP1B, RPTPe, and TCPTP, have been reported to promote obesity[5–12], supporting the notion that JAK2 inhibitory molecules increase risk for leptin resistance, obesity, and metabolic disease. Interestingly, we identified a JAK2-binding protein Sh2b1 as a potent positive regulator of JAK2 (refs. [13,14]); however, JAK2/Sh2b1 pathways in LepR-expressing neurons has not been explored in vivo.

Sh2b1 is an SH2 and PH domain-containing adaptor protein. It binds to JAK2 via its SH2 domain and robustly enhances JAK2 kinase activity[14,15]. Sh2b1 also binds to IRS1 and IRS2 and links JAK2 to IRS1/2-mediated activation of the PI 3-kinase pathway in cell cultures[13]. Aside from JAK2, Sh2b1 also binds to receptor tyrosine kinases, including insulin receptors, platelet-derived growth factor receptors, nerve growth factor receptor TrkA, and brain-derived neurotrophic factor (BDNF) receptor TrkB[16–22]. To examine Sh2b1 function in vivo, we generate and characterize global Sh2b1 knockout mice. Sh2b1-null mice develop severe leptin resistance, obesity, and type 2 diabetes[23,24]. We and other groups further demonstrate that the metabolic function of Sh2b1 has conserved from flies to humans. Deletion of Sh2b results in fat accumulation in Drosophila[25]. In human genome-wide association studies, numerous SH2B1 single-nucleotide polymorphisms (SNPs) have been identified to link to obesity, type 2 diabetes, and cardiovascular diseases[26–28]. Deletion of chromosomal 16p11.2, which encompasses the SH2B1 gene, is associated with severe obesity in humans[29–32]. Human SH2B1 missense mutations are cosegregated with the obesity and metabolic disease traits[33–38]. Thus, SH2B1 is emerging as a critical regulator of body weight and metabolism in both animals and humans; however, SH2B1 target cell types remain poorly understood.

We report that neuron-specific restoration of Sh2b1 expression reverses the obesity phenotypes of Sh2b1-null mice[39], indicating that neurons mediate Sh2b1 actions on body weight and metabolism. Given that Sh2b1 augments LepRb/JAK2 signaling in cell cultures, we postulate that Sh2b1 might cell-autonomously increase the ability of LepR neurons to control energy balance and body weight, perhaps by directly enhancing leptin signaling. In this study, we generate and characterize LepR cell-specific Sh2b1 knockout ($Sh2b1^{\Delta LepR}$) mice. $Sh2b1^{\Delta LepR}$ mice, like global Sh2b1 knockout mice, develop obesity, insulin resistance, and liver steatosis. Remarkably, Sh2b1 deficiency in LepR neurons abrogates the ability of leptin to stimulate sympathetic nerves innervating brown adipose tissue (BAT), leading to BAT dysfunction and reduced core body temperature in $Sh2b1^{\Delta LepR}$ mice. Collectively, our results unveil an unrecognized leptin/Sh2b1/ sympathetic nerve/adipose thermogenesis axis that combats obesity, type 2 diabetes, and liver steatosis.

## Results

**$Sh2b1^{\Delta LepR}$ mice spontaneously develop obesity.** To determine the role of Sh2b1 in LepR neurons, we generated $Sh2b1^{\Delta LepR}$ mice ($Sh2b1^{f/f};LepR$-$Cre^{+/+}$) by crossing $Sh2b1^{f/f}$ mice with $LepR$-$Cre$ drivers. $LepR$-$Cre$ mice were characterized previously[40,41]. Mice

were in a C57BL/6J background and fed a standard chow diet. $Sh2b1^{\Delta LepR}$ male and female mice progressively became heavier than sex/age-matched $Sh2b1^{f/f}$ and $LepR$-$Cre$ mice (Fig. 1a). Fat content was dramatically higher in $Sh2b1^{\Delta LepR}$ males and females relative to sex/age-matched $Sh2b1^{f/f}$ and $LepR$-$Cre$ mice (Fig. 1b). Both gonadal and inguinal white adipose tissue (WAT) depots were significantly larger in $Sh2b1^{\Delta LepR}$ relative to $Sh2b1^{f/f}$ and $LepR$-$Cre$ mice (Supplementary Fig. 1a). Individual white adipocyte size was substantially larger in $Sh2b1^{\Delta LepR}$ than in $LepR$-$Cre$ mice (Fig. 1c). Lean mass was not significantly different between $Sh2b1^{\Delta LepR}$ and $LepR$-$Cre$ mice (Supplementary Fig. 1b). To gain insight into the underlying mechanism, we analyzed energy balance. Food intake was relatively normal (Fig. 1d). $O_2$ consumption and $CO_2$ production (per mouse) were also not significantly different between $Sh2b1^{\Delta LepR}$ and $Sh2b1^{f/f}$ mice (Fig. 1e, Supplementary Fig. 1c). Of note, $O_2$ consumption and $CO_2$ production, after normalization to body weight, were significantly lower in $Sh2b1^{\Delta LepR}$ males and females relative to sex/age-matched $Sh2b1^{f/f}$ mice (Supplementary Fig. 1d, e). Core body temperature was significantly lower in $Sh2b1^{\Delta LepR}$ males and females relative to sex/age-matched $LepR$-$Cre$ or $Sh2b1^{f/f}$ mice (Fig. 1f, g). We further confirmed that $Sh2b1^{\Delta LepR}$ mice had lower body temperature using E-Mitters, and $Sh2b1^{\Delta LepR}$ locomotor activity was relatively normal (Fig. 1h). These data indicate that Sh2b1 in LepR neurons is indispensable for the maintenance of both body weight and core body temperature.

**$Sh2b1^{\Delta LepR}$ mice develop insulin resistance and liver steatosis.** Obesity promotes type 2 diabetes and nonalcoholic fatty liver disease (NAFLD), prompting us to assess insulin sensitivity and hepatic lipid content. $Sh2b1^{\Delta LepR}$ males and females developed hyperglycemia and hyperinsulinemia compared to sex/age-matched $LepR$-$Cre$ mice at 19–20 weeks of age (Fig. 2a). In glucose (GTT) or insulin (ITT) tolerance tests, blood glucose levels were markedly higher in $Sh2b1^{\Delta LepR}$ males and females relative to sex/ age-matched $LepR$-$Cre$ or $Sh2b1^{f/f}$ mice (Fig. 2b). Consistently, insulin-stimulated phosphorylation of Akt (pThr308 and pSer473) in liver and skeletal muscle was substantially lower in $Sh2b1^{\Delta LepR}$ than in $LepR$-$Cre$ mice (Fig. 2c). $Sh2b1^{\Delta LepR}$ mice also developed severe liver steatosis, as demonstrated by markedly increased lipid droplet number and size and triacylglycerol (TAG) levels in the liver (Fig. 2d, e). These results suggest that Sh2b1 in LepR neurons combats against insulin resistance, type 2 diabetes, and NAFLD.

**Adult-onset ablation of hypothalamic Sh2b1 results in obesity.** We recently reported that neuronal Sh2b1 promotes brain development[16]. To distinguish between brain development-dependent and -independent actions of Sh2b1 on body weight and metabolism, we generated adult-onset, hypothalamus-specific Sh2b1 knockout mice by bilaterally microinjecting AAV1-hSyn-Cre vectors into the mediobasal hypothalami (MBH) of $Sh2b1^{f/f}$ males at 10 weeks of age. AAV1-hSyn-green fluorescent protein (GFP) vectors were used as control. Bilateral MBH injections were histologically verified (Supplementary Fig. 2a). MBH-specific ablation of Sh2b1 substantially increased body weight and fat content (Fig. 3a, b). As an additional control, AAV1-hSyn-Cre vectors were bilaterally injected into the MBH of wild-type C57BL/6 males. There was no difference in body weight and fat content between the AAV1-hSyn-Cre and AAV1-hSyn-GFP groups (Supplementary Fig. 2b–c). $O_2$ consumption and $CO_2$ production (per mouse) were not significantly different between the AAV1-hSyn-Cre and AAV1-hSyn-GFP groups (Fig. 3c). Nonetheless, $O_2$ consumption and $CO_2$ production, after normalization to body weight, were significantly lower in AAV1-hSyn-Cre relative to AAV1-hSyn-GFP groups in the dark

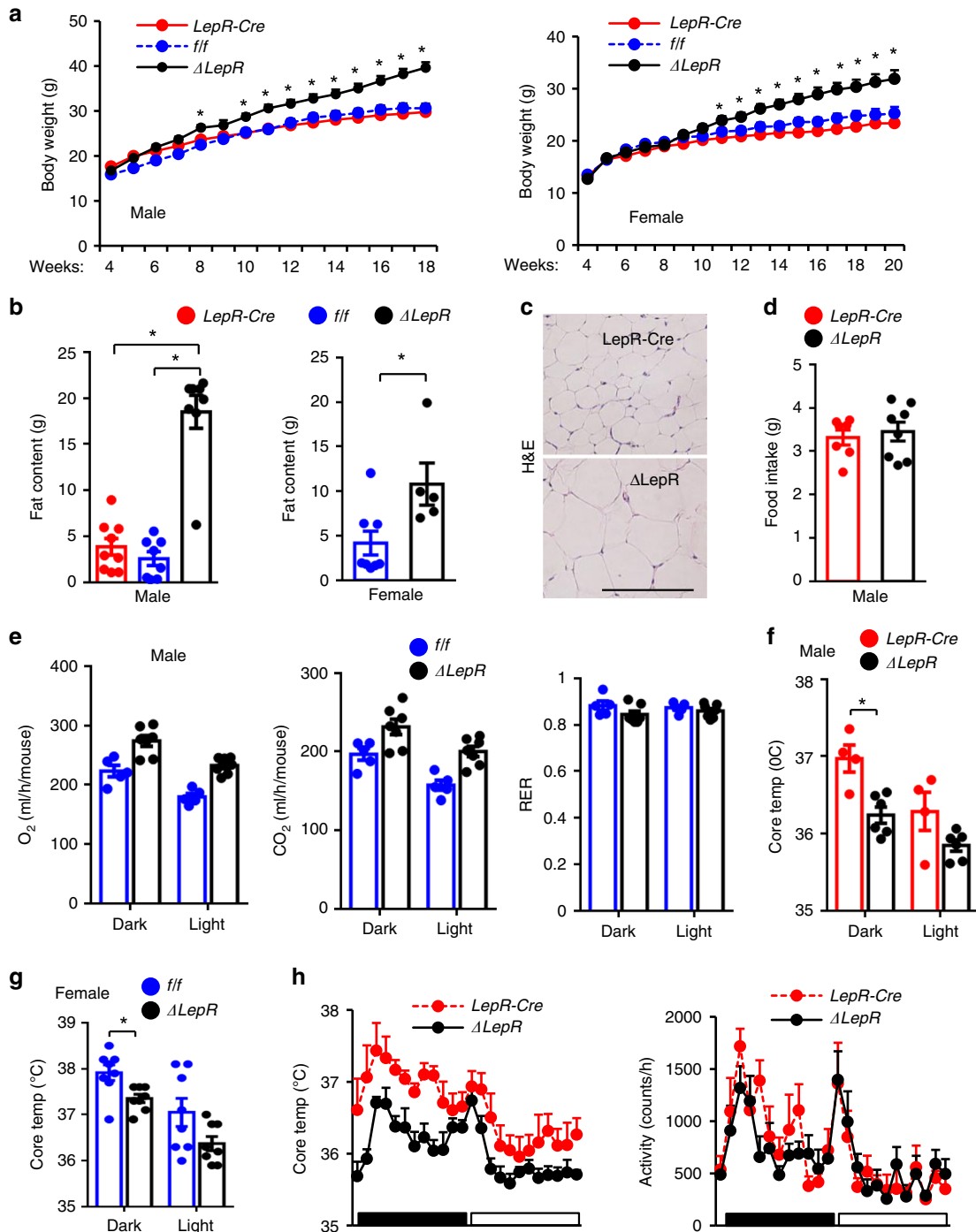

**Fig. 1 Sh2b1^{ΔLepR} mice develop obesity. a** Growth curves. Male: *f/f*: n = 10, *LepR-Cre*: n = 19, *Sh2b1^{ΔLepR}*: n = 25; female: *f/f*: n = 8, *LepR-Cre*: n = 15, *Sh2b1^{ΔLepR}*: n = 16. **b** Fat content. Male (22 weeks): *f/f*: n = 8, *Cre*: n = 9, *Sh2b1^{ΔLepR}*: n = 8; females (20 weeks): *f/f*: n = 8, *Sh2b1^{ΔLepR}*: n = 5. **c** Representative H&E staining of epididymal WAT sections at 22 weeks of age (3 pairs). Scale bar: 200 μm. **d** Food intake of males at 10 weeks of age. *LepR-Cre*: n = 7, *Sh2b1^{ΔLepR}*: n = 8. **e** $O_2$ consumption and $CO_2$ production at 10 weeks of age. Male: *f/f*: n = 5, *Sh2b1^{ΔLepR}*: n = 7. **f–g** Rectal temperature at 20 weeks of age. Male: *Cre*: n = 4, *Sh2b1^{ΔLepR}*: n = 6; female: *f/f*: n = 8, *Sh2b1^{ΔLepR}*: n = 8. **h** Core body temperature and locomotor activity in male mice (10 weeks) recorded using pre-implanted E-Mitters. *LepR-Cre*: n = 4, *Sh2b1^{ΔLepR}*: n = 6. Data are presented as mean ± SEM. *$p < 0.05$, two-tailed unpaired Student's *t*-test (**b** female, **d–g**), two-way ANOVA (**a**), or one-way ANOVA (**b** male). Source data are provided as a Source Data file.

phase (Supplementary Fig. 2d). Core body temperature was significantly lower in the AAV1-hSyn-Cre mice in the dark cycle (Fig. 3d). Notably, MBH-specific ablation of Sh2b1 significantly increased food intake (Fig. 3e). Like *Sh2b1^{ΔLepR}* mice, adult-onset and MBH-specific *Sh2b1* knockout mice developed hyperinsulinemia, glucose intolerance, and insulin resistance (Fig. 3f, g).

Insulin-stimulated phosphorylation of hepatic Akt was lower in AAV1-hSyn-Cre-transduced than in AAV1-hSyn-GFP-transduced *Sh2b1^{f/f}* mice (Fig. 3h). MBH-specific *Sh2b1* knockout mice also developed severe liver steatosis, as demonstrated by elevated levels of hepatocyte lipid droplets (Oil Red O staining of liver sections) and high levels of liver TAG (Fig. 3i). These data indicate that Sh2b1 in

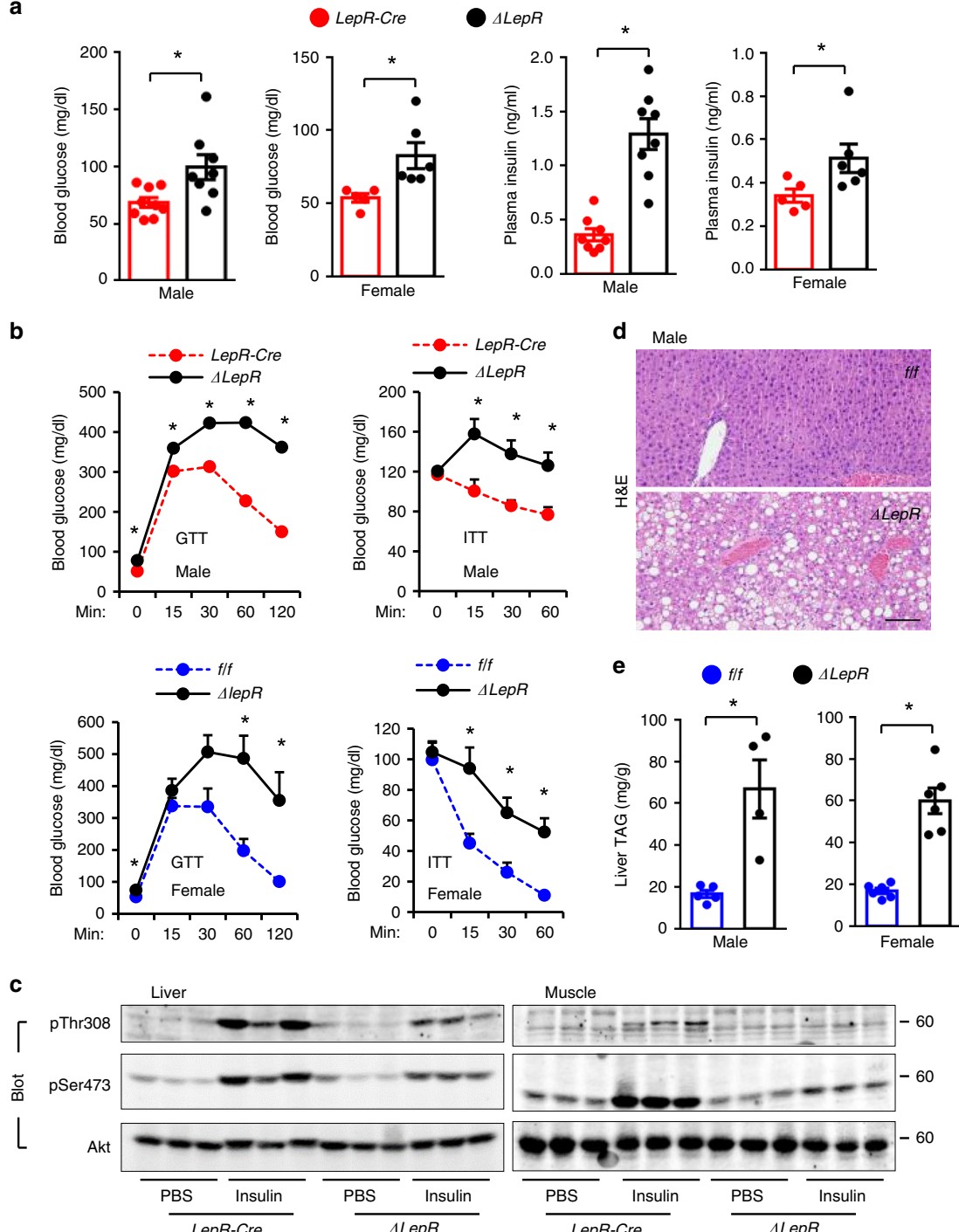

**Fig. 2 $Sh2b1^{\Delta LepR}$ mice develop insulin resistance, glucose intolerance, and liver steatosis. a** Overnight fasting blood glucose and insulin levels at 22 weeks of age. Male: $LepR$-$Cre$: $n = 8$ (insulin) and $n = 9$ (glucose), $Sh2b1^{\Delta LepR}$: $n = 8$; female: $LepR$-$Cre$: $n = 5$, $Sh2b1^{\Delta LepR}$: $n = 6$. **b** GTT and ITT at 20 weeks of age. Male: $LepR$-$Cre$: $n = 9$, $Sh2b1^{\Delta LepR}$: $n = 8$; female: $f/f$: $n = 5$, $Sh2b1^{\Delta LepR}$: $n = 5$. **c** Male mice (23 weeks) were stimulated with insulin. Liver and skeletal muscle extracts were immunoblotted with the indicated antibodies. Male: $f/f$: $n = 6$, $Sh2b1^{\Delta LepR}$: $n = 6$; female: $f/f$: $n = 6$, $Sh2b1^{\Delta LepR}$: $n = 6$. **d** Representative H&E staining of liver sections at 23 weeks of age (3 pairs). Scale bar: 200 μm. **e** Liver TAG levels (normalized to liver weight) at 23 weeks of age. Male: $f/f$: $n = 5$, $Sh2b1^{\Delta LepR}$: $n = 4$; female: $f/f$: $n = 6$, $Sh2b1^{\Delta LepR}$: $n = 6$. Data are presented as mean ± SEM. *$p < 0.05$, two-tailed unpaired Student's $t$-test. Source data are provided as a Source Data file.

the MBH regulates body weight, metabolism, food intake, and/or body temperature independently of its action on brain development.

**Hypothalamic overexpression of SH2B1 ameliorates obesity.** To determine whether MBH-specific overexpression of human SH2B1 protects against obesity, AAV9-CAG-SH2B1 or AAV9-

CAG-GFP (control) vectors were bilaterally injected into the MBH of C57BL/6J males. Mice were fed an HFD to induce obesity. Recombinant SH2B1 was detected in AAV9-CAG-SH2B1-transduced but not AAV9-CAG-GFP-transduced mice (Supplementary Fig. 3a, b). Body weight and fat content were significantly lower in the AAV9-CAG-SH2B1 group relative to

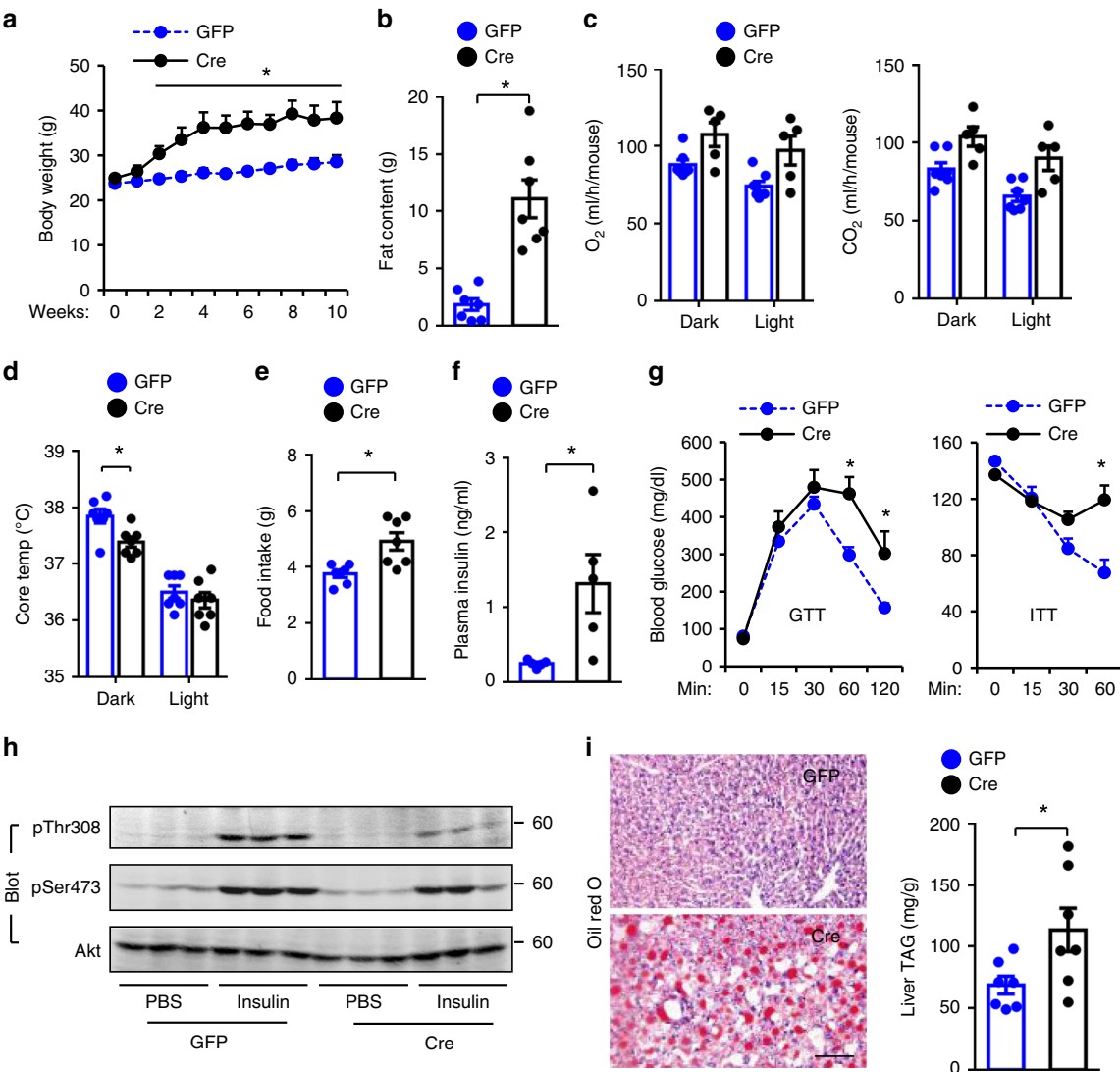

**Fig. 3 Adult-onset, MBH-specific ablation of Sh2b1 results in obesity, insulin resistance, and liver steatosis.** AAV1-hSyn-Cre or AAV1-hSyn-GFP vectors were bilaterally injected into the MBH of $Sh2b1^{f/f}$ males at 10 weeks of age. **a** Growth curves. **b** Fat content in 10 weeks post transduction. **c** Energy expenditure in 10 weeks after transduction. **d** Rectal temperature in 10 weeks following transduction. **e** Food intake at 10 weeks post transduction. **f** Overnight fasting plasma insulin levels in 10 weeks after transduction. **g** GTT and ITT in 10–11 weeks post transduction. **h** Mice were stimulated with insulin in 11 weeks post transduction. Liver extracts were immunoblotted with the indicated antibodies ($n = 3$ mice per group). **i** Representative Oil Red O staining of liver sections and liver TAG levels (normalized to liver weight) in 11 weeks post transduction. **a, b, d, e, g, i**: AAV1-hSyn-GFP: $n = 7$, AAV1-hSyn-Cre: $n = 7$; **c**: AAV1-hSyn-GFP: $n = 7$, AAV1-hSyn-Cre: $n = 5$; **f**: AAV1-hSyn-GFP: $n = 5$, AAV1-hSyn-Cre: $n = 5$. Scale bar: 100 μm. Data are presented as mean ± SEM. *$p < 0.05$, two-tailed unpaired Student's $t$-test. Source data are provided as a Source Data file.

the AAV9-CAG-GFP group (Fig. 4a, b). $O_2$ consumption and $CO_2$ production (per mouse) were not significantly different between the GFP and SH2B1 groups (Fig. 4c). Upon normalization to body weight, $O_2$ consumption and $CO_2$ production were significantly higher in the AAV9-CAG-SH2B1 group relative to the AAV9-CAG-GFP group (Supplementary Fig. 3c). Core body temperature was significantly higher in the AAV9-CAG-SH2B1 group (Fig. 4d). MBH-specific overexpression of SH2B1 substantially ameliorated HFD-induced insulin resistance and glucose intolerance, as assessed by ITT, GTT, and insulin-stimulated phosphorylation of Akt (Fig. 4e, f). Overexpression of SH2B1 in the hypothalamus also blocked HFD-induced liver steatosis, as demonstrated by a marked reduction in lipid droplet number and size and liver TAG levels in the AAV9-CAG-SH2B1 relative to AAV9-CAG-GFP groups (Fig. 4g). These data further confirm that hypothalamic Sh2b1,

perhaps in LepR neurons, protects against obesity, type 2 diabetes, and NAFLD in adult mice.

**LepR neuron Sh2b1 is required for brown fat thermogenesis.** Given that BAT and beige fat promote adaptive thermogenesis, energy expenditure, and weight loss, we examined the impact of hypothalamic Sh2b1 on BAT activity. Ablation of Sh2b1 in either LepR neurons ($Sh2b1^{\Delta LepR}$ mice) or the MBH (AAV1-hSyn-Cre-transduced $Sh2b1^{f/f}$ mice) caused whitening of BAT (e.g. enlarged lipid droplets) and dramatic downregulation of uncoupling protein 1 (Ucp1) (Fig. 5a). Ucp1 protein and mRNA were barely detectable in $Sh2b1^{\Delta LepR}$ mice at 22 weeks of age (Fig. 5b, c). Ucp1 expression in inguinal WAT markedly decreased in $Sh2b1^{\Delta LepR}$ mice (Supplementary Fig. 1g). Of note, absolute expression levels of Ucp1 was markedly higher in BAT than in WAT. Conversely, MBH-specific overexpression of SH2B1 reversed HFD-induced

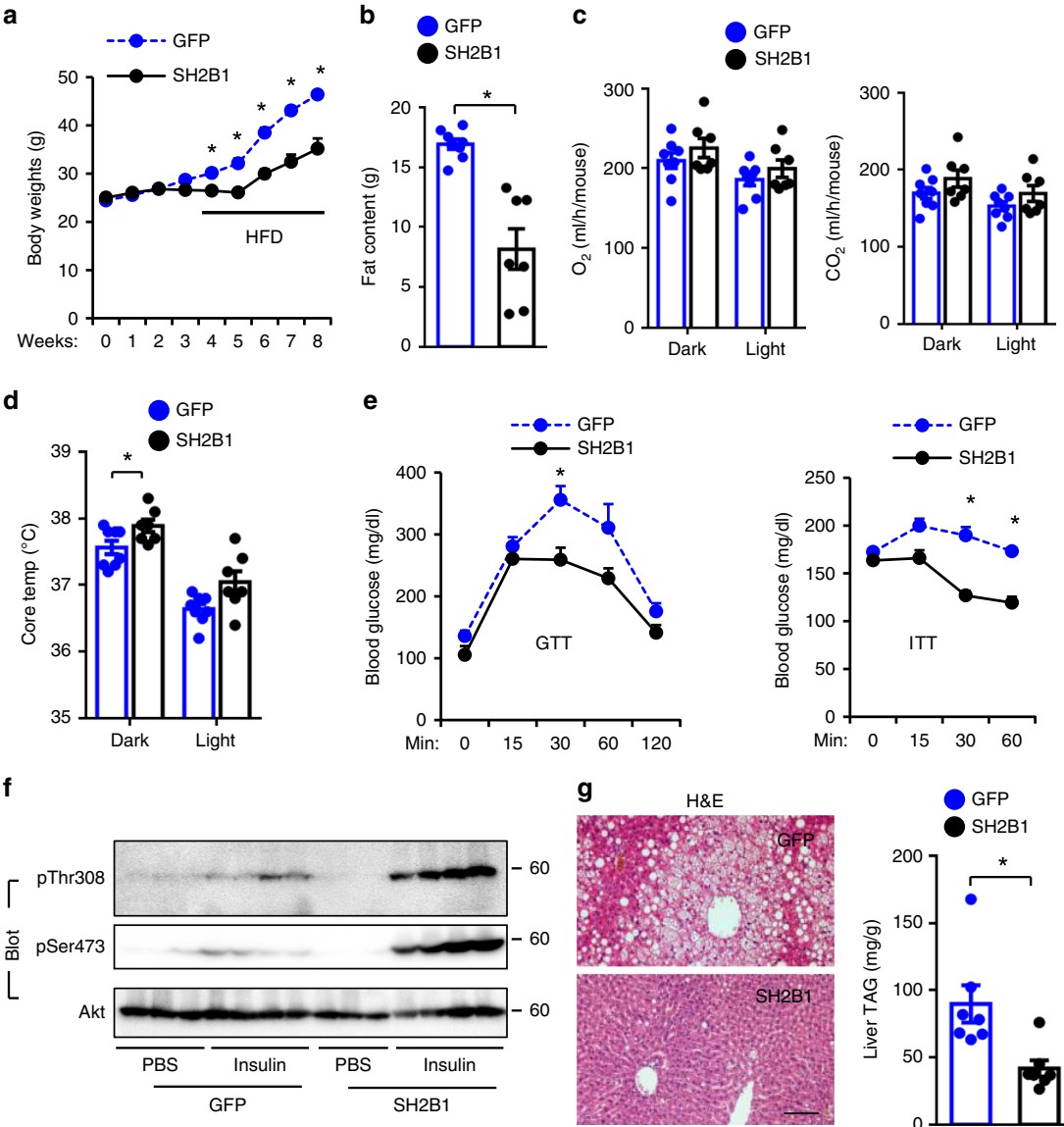

**Fig. 4 MBH-specific overexpression of SH2B1 protects against HFD-induced obesity, insulin resistance, and liver steatosis.** AAV9-CAG-SH2B1β ($n =$ 7) or AAV9-CAG-GFP ($n = 7$–8) vectors were bilaterally injected into the MBH of adult C57BL/6J males, followed by HFD feeding. **a** Growth curves. **b** Fat content in 10 weeks post HFD. **c** Energy expenditure in 10 weeks post HFD. **d** Rectal temperature in 10 weeks post HFD. **e** GTT and ITT in 10–11 weeks post HFD. **f** Mice (11 weeks post HFD) were stimulated with PBS ($n = 3$ mice per group) or insulin ($n = 4$ mice per group). Liver extracts were immunoblotted with the indicated antibodies. **g** Representative H&E staining of liver sections and liver TAG levels (normalized to liver weight) in 12 weeks post HFD. Scale bar: 200 μm. Data are presented as mean ± SEM. *$p < 0.05$, two-tailed unpaired Student's $t$-test. Source data are provided as a Source Data file.

whitening of BAT and downregulation of Ucp1 (Fig. 5a). Ucp1 protein and mRNA levels were substantially higher in AAV9-CAG-SH2B1-transduced relative to AAV9-CAG-GFP-transduced mice (Fig. 5b, c). Ucp1 mediates adaptive thermogenesis and energy expenditure[42]. Accordingly, $Sh2b1^{\Delta LepR}$ mice displayed markedly lower core body temperature compared to $LepR$-$Cre$ mice upon cold exposure (on a chow diet), and mice with MBH-specific overexpression of SH2B1 (on a HFD) had the opposite effects (Fig. 5d). Thus, we uncovered an unrecognized hypothalamic Sh2b1/BAT axis that critically regulates adaptive thermogenesis and core body temperature.

**LepR neuron Sh2b1 mediates leptin stimulation of the SNS.** Considering the pivotal role of the sympathetic nervous system (SNS) in BAT activation, we assessed the impact of Sh2b1

deficiency on the ability of leptin to stimulate sympathetic nerve transmissions in BAT. Both male and female $Sh2b1^{\Delta LepR}$ mice developed hyperleptinemia (Fig. 6a). Leptin stimulated phosphorylation of hypothalamic Stat3 to a lower degree in $Sh2b1^{\Delta LepR}$ relative to $Sh2b1^{f/f}$ mice (Fig. 6b, c). Conversely, MBH-specific overexpression of SH2B1 augmented leptin-stimulated phosphorylation of hypothalamic Stat3 (Fig. 6b). To directly assess sympathetic nerve activity (SNA), we electrophysiologically recorded BAT SNA in $Sh2b1^{\Delta LepR}$ mice at both 6 weeks (prior to the onset of obesity) and 12 weeks of age. Baseline SNA was significantly lower in $Sh2b1^{\Delta LepR}$ relative to $Sh2b1^{f/f}$ mice (Fig. 6d, e). In agreement with the previous reports[43], central injection of leptin progressively and markedly increased SNA in wild type (i.e. $Sh2b1^{f/f}$) mice (Fig. 6d, f). It is likely that multiple synaptic modifications and/or polysynaptic transmissions contribute to a delayed onset of the leptin action on SNA. Strikingly, deletion of

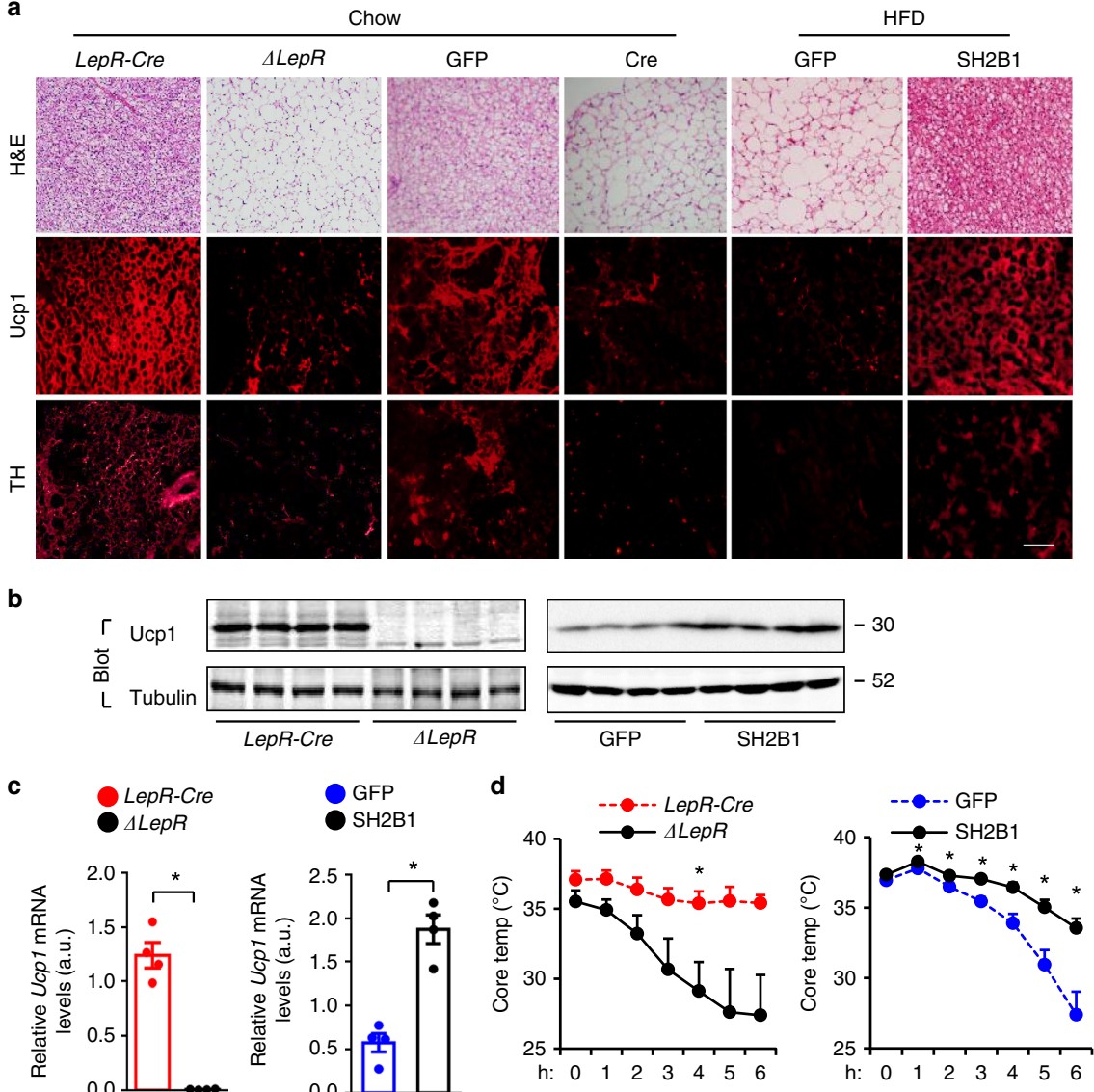

**Fig. 5 Sh2b1 in LepR neurons supports the maintenance and function of BAT. a–c** BAT was harvested from $Sh2b1^{\Delta LepR}$ and $LepR$-$Cre$ males at 10 weeks of age, and from $Sh2b1^{f/f}$ males in 11 weeks after they were bilaterally injected into the MBH with AAV1-hSyn-Cre or AAV1-hSyn-GFP vectors (on chow diet). Adult C57BL/6J males were bilaterally injected into the MBH with AAV9-CAG-SH2B1β or AAV9-CAG-GFP vectors and then fed an HFD for 12 weeks. **a** Representative BAT images. Scale bar: 200 μm. $Sh2b1^{\Delta LepR}$: $n = 3$, $LepR$-$Cre$: $n = 4$, AAV1-hSyn-Cre: $n = 7$, AAV1-hSyn-GFP: $n = 7$, AAV9-CAG-GFP: $n = 8$, AAV9-CAG-SH2B1: $n = 8$. **b** BAT extracts were immunoblotted with the indicated antibodies. $Sh2b1^{\Delta LepR}$: $n = 4$, $LepR$-$Cre$: $n = 4$, AAV9-CAG-GFP: $n = 4$, AAV9-CAG-SH2B1: $n = 4$. **c** Ucp1 mRNA levels were normalized to 36B4 levels. $Sh2b1^{\Delta LepR}$: $n = 4$, $LepR$-$Cre$: $n = 4$, AAV9-CAG-GFP: $n = 4$, AAV9-CAG-SH2B1β: $n = 4$. a.u. arbitrary units. **d** Core body temperature was recorded using E-Mitters in $Sh2b1^{\Delta LepR}$ ($n = 3$) and $LepR$-$Cre$ ($n = 3$) at 10 weeks of age, on chow diet, and upon cold temperature (4 °C). Adult C57BL/6J males were bilaterally injected into the MBH with AAV9-CAG-GFP ($n = 7$) or AAV9-CAG-SH2B1β ($n = 7$) vectors and then fed an HFD for 10 weeks. Rectal temperature was measured upon cold exposure. Data are presented as mean ± SEM. *$p < 0.05$, two-tailed unpaired Student's $t$-test. Source data are provided as a Source Data file.

$Sh2b1$ in LepR neurons completely abrogated the ability of leptin to stimulate SNA in $Sh2b1^{\Delta LepR}$ mice (Fig. 6f). These results indicate the Sh2b1 branch of LepR signaling pathways is required for leptin to stimulate the SNS.

We next set out to examine neuronal activity (c-Fos expression as a surrogate marker) in the central sympathetic network, focusing on the POA, DMH, ARC, paraventricular hypothalamus (PVH), and rostral raphe pallidus (rRPa). These regions are known to control sympathetic outflows to BAT[44]. Cold exposure rapidly and robustly increased the number of c-Fos neurons in $Sh2b1^{f/f}$ mice (Fig. 6g, h). Sh2b1 deficiency substantially suppressed cold-stimulated neuronal activation in the POA, DMH, and rRPa of $Sh2b1^{\Delta LepR}$ mice (Fig. 6g,

h). In the ARC, neural activity was significantly lower in $Sh2b1^{\Delta LepR}$ relative to $Sh2b1^{f/f}$ mice at both 22 and 4 °C (Fig. 6h).

To further confirm the role of LepR neuron Sh2b1 in regulating the SNS, we measured the levels of tyrosine hydroxylase (TH), a sympathetic nerve marker, in BAT. We previously validated that anti-TH antibody specifically recognizes TH in immunostaining[45]. Immunoreactivity to TH was dramatically lower in $Sh2b1^{\Delta LepR}$ relative to $LepR$-$Cre$ mice (Fig. 5a). Likewise, TH levels in BAT were also markedly reduced by MBH-specific ablation of Sh2b1 (AAV-Cre vs AAV-GFP groups) (Fig. 5a). Conversely, MBH-specific overexpression of SH2B1 increased BAT TH levels (AAV-SH2B1 vs AAV-GFP) (Fig. 5a). Collectively, these results unveil an

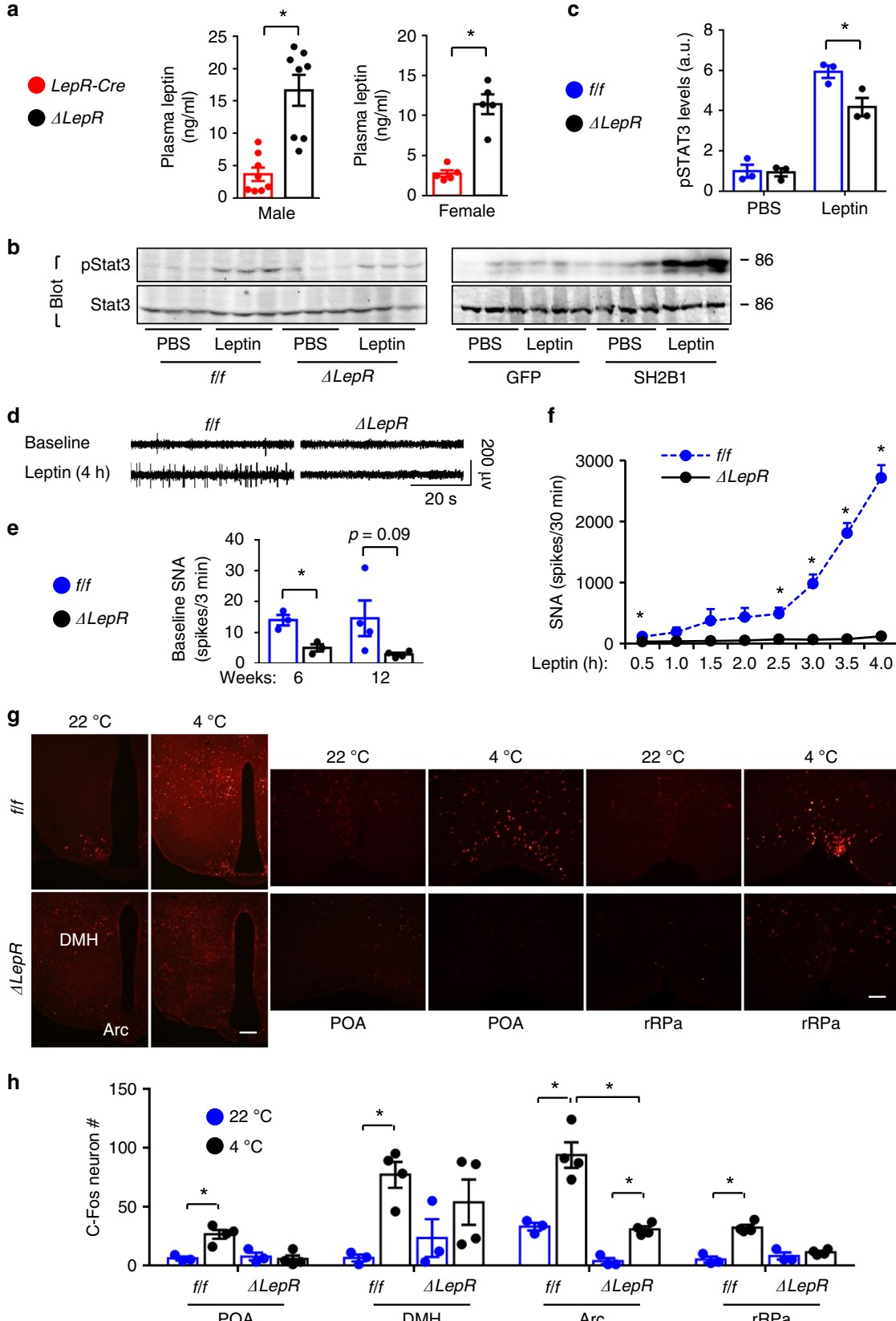

unrecognized Sh2b1/hypothalamic sympathetic network/SNS/BAT energy expenditure axis.

**LepR neuron Sh2b1 preserves adipose SNS integrity in aging.** $Sh2b1^{\Delta LepR}$ mice develop obesity in an age-dependent manner,

prompting us to examine age-associated SNS degeneration. BAT TH levels were normal in young $Sh2b1^{\Delta LepR}$ mice prior to 8 weeks of age (Fig. 7a, b), indicating that Sh2b1 in LepR neurons is dispensable for the development of adipose SNS. TH levels decreased progressively post 8 weeks of age and were barely

**Fig. 6 Sh2b1 in LepR neurons mediates leptin stimulation of adipose SNS. a** Overnight fasting plasma leptin levels at 22 weeks of age. Male: *LepR-Cre*: $n = 8$, *Sh2b1$^{\Delta LepR}$*: $n = 8$; female: *LepR-Cre*: $n = 5$, *Sh2b1$^{\Delta LepR}$*: $n = 5$. **b**, **c** *Sh2b1$^{\Delta LepR}$* and *Sh2b1$^{f/f}$* males (6 weeks) were treated with leptin. Hypothalamic extracts were immunoblotted with antibodies against phospho-Stat3 (pTyr705) or Stat3. Stat3 phosphorylation was normalized to total Stat3 levels. Adult C57BL/6J males were bilaterally injected into the MBH with AAV9-CAG-SH2B1β or AAV9-CAG-GFP vectors, fed an HFD for 10 weeks, fasted overnight, and centrally injected with leptin for 15 min. Phosphorylation of hypothalamic Stat3 was assessed. **c** $n = 3$ per group. a.u. arbitrary units. **d–f** BAT sympathetic nerve activity (SNA). **d** Representative traces. **e** Baseline SNA at 6 ($n = 3$ mice per group) and 12 ($n = 4$ mice per group) weeks of age. **f** Time courses of leptin response. Leptin was centrally administrated into *Sh2b1$^{\Delta LepR}$* ($n = 3$) and *Sh2b1$^{f/f}$* ($n = 3$) males at 6 weeks of age. **g**, **h** *Sh2b1$^{\Delta LepR}$* and *Sh2b1$^{f/f}$* males (7 weeks) were exposed to 22 or 4 °C for 4 h. Hypothalamic sections were stained with anti-Fos antibody. **g** Representative c-Fos staining of hypothalamic sections. Scale bar: 200 μm. **h** Number of c-Fos neurons in different hypothalamic areas. 4 °C: $n = 4$ mice per group, 22 °C: $n = 3$ mice per group. Data are presented as mean ± SEM. *$p < 0.05$, two-tailed unpaired Student's *t*-test (**a**, **c**, **e**, **f**) or two-way ANOVA (**h**). Source data are provided as a Source Data file.

detectable in *Sh2b1$^{\Delta LepR}$* mice at 22 weeks of age (Fig. 7a, b). To confirm sympathetic degeneration, we assessed the levels of class III β-tubulin, a neuronal marker, using antibody TUJ1. TUJ1 immunoreactivity in BAT was abundant in *Sh2b1$^{f/f}$* but not *Sh2b1$^{\Delta LepR}$* mice at 22 weeks of age (Fig. 7c). Of note, Ucp1 downregulation followed the course of SNS deterioration in *Sh2b1$^{\Delta LepR}$* mice (Fig. 7d).

Next, we asked whether Sh2b1 deficiency in LepR neurons worsens SNS degeneration in white adipose tissue (WAT). Because sympathetic innervation of WAT is sparse and difficult to be detected[46,47], we assessed phosphorylation of hormone-sensitive lipase (HSL), a surrogate marker for SNS activation. HSL phosphorylation (pSer563 and pSer660) was normal or slightly higher in *Sh2b1$^{\Delta LepR}$* mice at 3 weeks of age; thereafter, HSL phosphorylation decreased progressively and became barely detectable at 22 weeks of age (Fig. 7e). Collectively, these results suggest that Sh2b1 in LepR neurons is involved in preserving the SNS in both BAT and WAT during aging.

**Deletion of POMC neuron *Sh2b1* is unable to induce obesity.** We next aimed to further map Sh2b1 target neurons. Brain sections were prepared from wild type and global *Sh2b1* knockout (negative control) mice and stained with anti-Sh2b1 antibody. Sh2b1 was detected in hypothalamic cells in wild-type mice (Supplementary Fig. 4a). To confirm these results, we generated *Sh2b1-Cre* knockin mice by inserting an *IRES-eGFP-2A-Cre* cassette into the *Sh2b1* locus 3′ to the STOP codon (Supplementary Fig. 4b). GFP levels in *Sh2b1-Cre* mice were below detection thresholds. To facilitate detection of Sh2b1 neurons, *Sh2b1-Cre* drivers were crossed with *Rosa-mTmG* reporter mice to genetically label Sh2b1 neurons with mGFP in *Sh2b1-Cre;Rosa-mTmG* mice. We found that mGFP (a marker for expression of endogenous Sh2b1) was expressed in most of hypothalamic cells in *Sh2b1-Cre;Rosa-mTmG* but not *Rosa-mTmG* mice (Supplementary Fig. 4c). In line with these findings, Sh2b1 protein is detected in the entire brain by immunoblotting[16]. To confirm that pro-opiomelanocortin (POMC) and AgRP neurons express Sh2b1, hypothalamic sections were prepared from *Sh2b1-Cre;mTmG* mice and immunostained with antibodies to POMC and AgRP. Both POMC and AgRP neurons expressed mGFP (Supplementary Fig. 4d).

To explore the role of Sh2b1 in POMC neurons, we generated POMC neuron-specific *Sh2b1* knockout (*Sh2b1$^{\Delta POMC}$*) mice (*Sh2b1$^{f/f}$;POMC-Cre$^{+/-}$*) by crossing *Sh2b1$^{f/f}$* mice with *POMC-Cre* drivers. Unlike *Sh2b1$^{\Delta LepR}$* mice, *Sh2b1$^{\Delta POMC}$* mice were grossly normal when fed a chow diet (Supplementary Fig. 5a). We placed *Sh2b1$^{\Delta LepR}$* mice on HFD for 10 weeks. Body weight, fat content, GTT, and ITT were comparable between *Sh2b1$^{\Delta LepR}$*, *POMC-Cre*, and *Sh2b1$^{f/f}$* mice (Supplementary Fig. 5b–e). Together, these data suggest that POMC neuron-specific ablation of Sh2b1 is insufficient to induce obesity and metabolic disease.

## Discussion

We herein identify LepR neurons as key Sh2b1 targets that mediate Sh2b1 protection against obesity, type 2 diabetes, and NAFLD. We demonstrated that LepR neuron-specific deletion of *Sh2b1*, or adult-onset deletion of *Sh2b1* in the hypothalamus (containing LepR neurons), resulted in severe obesity, insulin resistance, and liver steatosis. Conversely, MBH-specific over-expression of SH2B1 ameliorated HFD-induced obesity and metabolic syndromes. Leptin stimulation of the hypothalamic JAK2/Stat3 pathway was impaired in *Sh2b1$^{\Delta LepR}$* mice, supporting the notion that Sh2b1 is an endogenous sensitizer for leptin action, perhaps by enhancing JAK2 activation. Remarkably, ablation of Sh2b1 in LepR neurons abrogated the ability of leptin to stimulate sympathetic nerves projecting to BAT. Likewise, adult-onset, MBH-specific ablation of Sh2b1 also impaired sympathetic transmissions in BAT. BAT became whitening and impaired in adaptive thermogenesis in both *Sh2b1$^{\Delta LepR}$* mice and mice with MBH-specific ablation of Sh2b1, presumably owing to adipose SNS-deficits. Consequently, core body temperature was low and cold tolerance was impaired in both *Sh2b1$^{\Delta LepR}$* mice and MBH-specific *Sh2b1* knockout mice. These findings define LepR neuron Sh2b1 as a critical central regulator of thermogenesis and body temperature. Thus, we unveil an unrecognized leptin/LepR neuron Sh2b1/SNS/BAT/thermogenesis/body temperature axis. However, we cannot exclude the possibility that hypothermic Sh2b1 may increase thermogenesis and body temperature by an additional leptin-independent mechanism. For instance, Sh2b1 may enhance the ability of interleukin-6, a well-known pyrogenic cytokine, to increase thermogenesis and body temperature through enhancing the JAK2/Stat3 pathway. Furthermore, hypothalamic Sh2b1 may increase body temperature by a SNS-independent mechanism, perhaps by enhancing the ability of hypothalamic–pituitary–thyroid axis to increase thermogenesis and body temperature. Given that the SNS/BAT pathway increases energy expenditure and weight loss, the LepR neuron Sh2b1/SNS/BAT/thermogenesis pathway is expected to mediate leptin stimulation of energy expenditure. Notably, food intake was relatively normal in *Sh2b1$^{\Delta LepR}$* mice, raising the possibility that leptin may regulate food intake and energy expenditure by Sh2b1-independent and Sh2b1-dependent pathways, respectively.

Both aging and obesity are associated with impaired adipose sympathetic nerve transmissions[48–50], but the underlying mechanism is poorly understood. We found that BAT sympathetic nerve fibers deteriorated age-dependently and became undetectable in *Sh2b1$^{\Delta LepR}$* mice after 22 weeks of age. Following the course of sympathetic nerve degeneration, BAT progressively lost Ucp1 expression and thermogenic capability. Age-associated sympathetic nerve degeneration also occurred in WAT in *Sh2b1$^{\Delta LepR}$* mice. Of note, BAT sympathetic innervation and transmission were normal in young *Sh2b1$^{\Delta LepR}$* mice prior to 8 weeks of age. These results suggest that LepR neuron Sh2b1 is

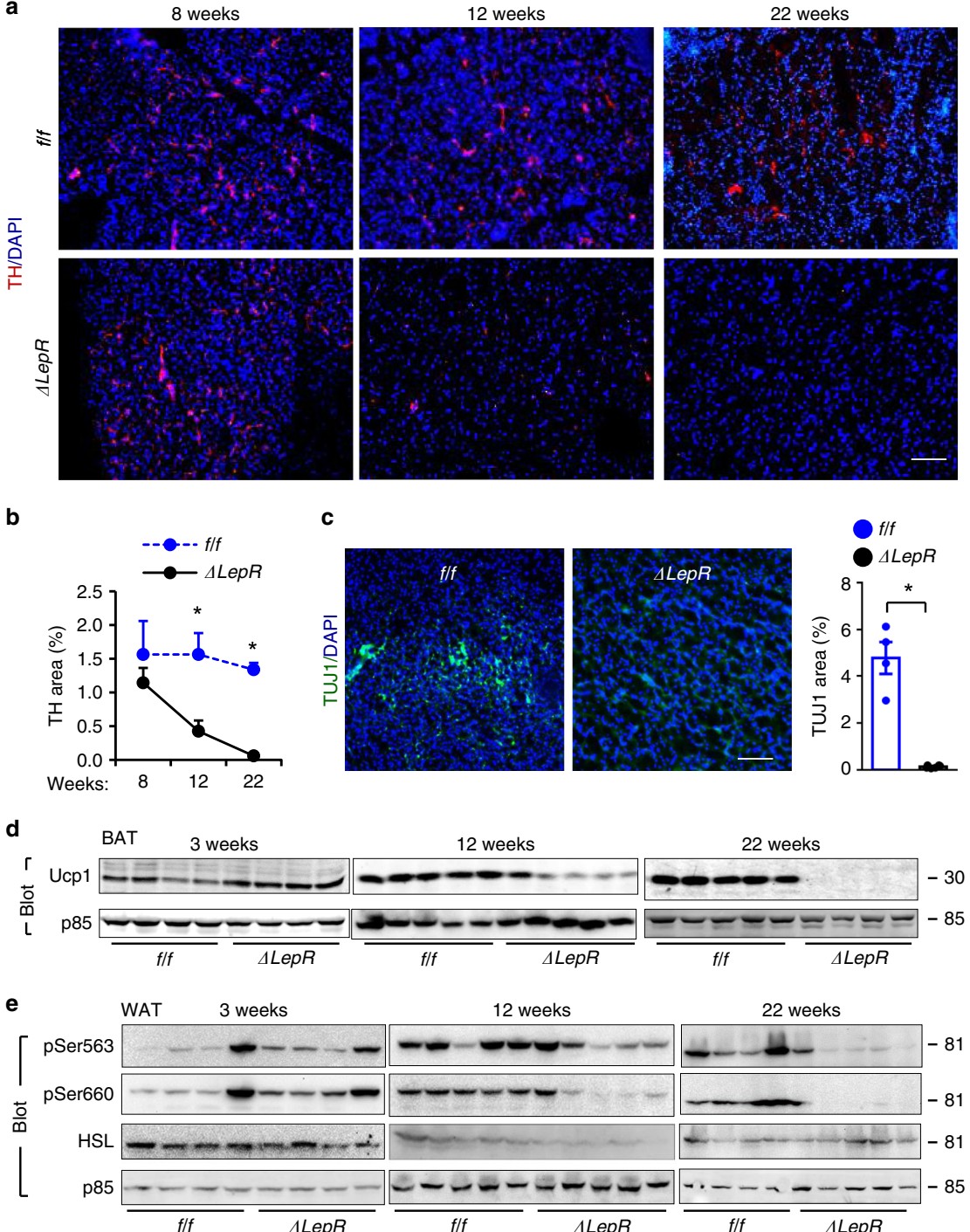

**Fig. 7 Sh2b1 in LepR neurons supports the maintenance of adipose SNS. a**, **b** BAT sections were prepared from male mice at 8, 12, and 22 weeks of age and stained with anti-tyrosine hydroxylase (TH) antibodies. **a** Representative images ($n = 3$ mice per group). **b** TH areas were quantified and normalized to total areas ($n = 3$ mice per group). **c** BAT sections were prepared at 22 weeks of age and stained with anti-TUJ1 antibody. TUJ1 areas were normalized to total areas. $Sh2b1^{f/f}$: $n = 4$, $Sh2b1^{\Delta LepR}$: $n = 6$. **d**, **e** BAT (**d**) and epididymal WAT (**e**) extracts were prepared from male mice at 3, 12, and 22 weeks of age and immunoblotted with the indicated antibodies. Each lane represents an individual mouse. Data are presented as mean ± SEM. *$p < 0.05$, two-tailed unpaired Student's $t$-test. Source data are provided as a Source Data file.

not required for adipose SNS development; rather, it plays an important role in preserving the adipose SNS against degeneration. Given that Sh2b1 is expressed broadly in the hypothalamus, it is not surprising that the neuronal activity of the central sympathetic network, particularly in the POA, DMH, and rRPa, is inhibited in $Sh2b1^{\Delta LepR}$ mice. These results suggest that hypothalamic Sh2b1 preserves adipose SNS integrity and sympathetic

transmissions by a top-down mechanism, perhaps by enhancing leptin and/or other hormone and neuropeptide signaling in the central sympathetic network. We acknowledge that our data do not exclude the possibility that adipose SNS deterioration may be secondary to obesity in $Sh2b1^{\Delta LepR}$ mice. Additional studies are warranted to further characterize hypothalamic Sh2b1 circuits that protect against adipose SNS degeneration.

The leptin resistance phenotype of $Sh2b1^{\Delta LepR}$ mice supports a concept that Sh2b1 is an endogenous enhancer of leptin sensitivity in vivo. Strikingly, ablation of Sh2b1 in LepR neurons abolished leptin stimulation of the adipose SNS, indicating that Sh2b1 mediates leptin actions on the SNS and energy expenditure. We acknowledge that we did not detect significant difference in energy expenditure (normalized to mice) between $Sh2b1^{\Delta LepR}$ and $Sh2b1^{f/f}$ mice. We also did not observe the RER circadian rhythm. We speculate that assay-related stress and/or other factors may influence energy expenditure and the circadian rhythm, thereby masking difference between these two groups. Notably, the hypothalamic PI 3-kinase pathway was reported to mediate leptin stimulation of the SNS[51,52]. Sh2b1 mediates leptin stimulation of the PI 3-kinase pathway by recruiting IRS proteins to JAK2[13], raising the possibility that the LepRb/Sh2b1/PI 3-kinase pathway may mediate leptin stimulation of the SNS/BAT/thermogenesis axis. However, we do not exclude the possibility that Sh2b1 in LepR neurons may promote SNS activity and maintenance by additional mechanisms. For instance, Sh2b1 enhances BDNF and insulin signaling in cell cultures[16–18,53]. BDNF and insulin, like leptin, also activate the SNS/BAT energy expenditure axis[16–18,54,55]. Sh2b1 may increase the ability of insulin and/or BDNF to stimulate energy expenditure. Of note, deletion of neither $Sh2b2$ nor $Sh2b3$ results in obesity[56,57], suggesting that the Sh2b1 actions on energy balance and body weight cannot be replaced by other Sh2b family members.

We found that Sh2b1 was widely expressed in the entire brain, including hypothalamic POMC neurons and AgRP neurons. $Sh2b1^{\Delta POMC}$ mice with POMC neuron-specific ablation of Sh2b1, unlike $Sh2b1^{\Delta LepR}$ mice, did not develop obesity and insulin resistance. These results are not unexpected, considering that ablation of LepR in POMC neurons, AgRP neurons, or both only slightly affects body weight and metabolism[58,59]. Ablation of LepR in Ghrh, Htr2c, or Prlh neurons also fails to cause obesity[60,61]. We postulate that Sh2b1 deficiency in POMC neurons may activate a compensatory mechanism in other Sh2b1 subpopulations, which masks the action of POMC neuron Sh2b1 on body weight and metabolism.

In conclusion, we unravel an unrecognized LepR neuron Sh2b1/SNS/adipose energy expenditure axis that combats against obesity, type 2 diabetes, and NAFLD. LepR neuron Sh2b1 mediates leptin stimulation of the SNS and supports preservation of adipose SNS against degeneration. The Sh2b1/SNS/fat axis may serve as a potential therapeutic target for the treatment of obesity and metabolic disease.

## Methods

**Animals**. $Sh2b1^{f/f}$ and $LepR$-$Cre$ (knockin of $Cre$ in the $LepR$ 3′UTR) mice (C57BL/6 background) were characterized previously[40,62]. $Sh2b1^{f/f}$ mice were crossed with $LepR$-$Cre$ mice to generate $Sh2b1^{\Delta LepR}$ mice ($Sh2b1^{f/f};Cre^{+/+}$). Because Cre levels in $LepR$-$Cre^{+/−}$ mice are insufficient to delete target genes[41,63,64], we generated homozygous $LepR$-$Cre^{+/+}$ mice to delete $Sh2b1$. A target vector containing an $IRES$-$EGFP$-$2A$-$Cre$ fragment at the $Sh2b1$ locus (3′ to the STOP codon) was used to generate $Sh2b1$-$Cre$ knockin mice through ES cell homologous recombination (Cyagen Biosciences Inc., Santa Clara, CA). $Rosa26$-$mTmG$ reporter mice were from the Jackson Laboratory (007676). Mice were housed on a 12 h light–dark cycle and fed ad libitum either a normal chow diet (9% fat; TestDiet, St. Louis, MO) or an HFD (60% fat; Research Diets, New Brunswick, NJ).

**Ethics statements**. Animal research was complied with all relevant ethnic regulations. Animal experiments were conducted following the protocols approved by the University of Michigan Institutional Animal Care and Use Committee (IACUC).

**Stereotaxic microinjection**. Mice were anesthetized with isoflurane and mounted on an Ultra Precise Small Animal Stereotaxic Alignment System (Model 963, David KOPF Instruments, Tujunga, CA). Body temperature was maintained within a normal range during the entire procedures using a thermal pad. The skull was exposed to identify the bregma and lambda, and a small opening in the skull was made using a bone drill at coordinates (mm): −1.5 (a–p), ±0.4 (m–l), and −5.8 (d–v). AAV vectors (0.5 μl) were injected into the hypothalamus in 10 min using UltraMicroPumps with SYS-Micro4 Controller (UMP3-2, World Precision Instruments Inc, Sarasota, FL). AAV1.hSyn.HI.eGFP-Cre and AAV1.hSyn.eGFP (control) were purchased from the Penn Vector Core, University of Pennsylvania School of Medicine. Cre expression is under the control of the neuron-specific human $synapsin$ promoter. We prepared AAV9-CAG-SH2B1 and AAV9-CAG-GFP (control) vectors in which SH2B1β expression is under the control of the constitutively active synthetic $CAG$ promoter.

**Body core temperature and locomotor activity**. Mice were anesthetized with isoflurane, and a G2 E-Mitter (870-0010-01, Starr life Sciences Corp, Oakmont, PA) was surgically implanted in the abdominal cavity. Locomotor activity and core body temperature were monitored using ER-4000 Energizer/Receiver (Bend, OR). Data were analyzed using Vitalview software (Starr life Sciences Corp, Oakmont, PA).

**Plasma insulin and leptin, GTT, and ITT**. Blood samples were collected from tail veins. Plasma insulin and leptin were measured using insulin and leptin ELISA kits (CRYSTAL CHEM, Downers Grove, IL), respectively. For GTT, mice were fasted for 16 h and intraperitoneally injected with glucose (2 g/kg body weight), and blood glucose was measured in 0, 15, 30, 60, and 120 min after injection. For ITT, mice were fasted for 6 h and intraperitoneally injected with insulin (0.7 U/kg for $Sh2b1^{\Delta LepR}$ mice, 0.6 U/kg for AAV-Cre or AAV-SH2B1β transduced mice), and blood glucose was measured in 0, 15, 30, and 60 min after injection.

**Fat content and energy expenditure**. Fat content and lean body mass were measured using a dual-energy X-ray absorptiometry pDexa (Norland Stratec). Heads, neck, and tails were not included in the pDexa analysis. Energy expenditure was measured by indirect calorimetry (Windows Oxymax Equal Flow System, Columbus Instruments, Columbus, OH).

**Body composition and liver TAG levels**. Mice were euthanized and organs were harvested and weighted. Liver samples were homogenized in 1% acetic acid and extracted using chloroform:methanol (2:1). The organic phase was dried via evaporation and dissolved in isopropanol. TAG levels were measured using a TAG assay kit (Pointe Scientific Inc., Canton, MI) and normalized to liver weight.

**Immunostaining**. Brain and BAT sections were prepared using a Leica cryostat (Leica Biosystems Nussloch GmbH, Nussloch, Germany), and immunostained with the indicated antibodies (Supplemental Table 1). Images were visualized using a BX51 Microscope (Olympus, Tokyo, Japan) and a DP72 digital camera (Olympus, Tokyo, Japan).

**Immunoblotting**. Mice were fasted overnight and injected with insulin, and liver and skeletal muscle were harvested 5 min later. $Sh2b1^{\Delta LepR}$ and $Sh2b1^{f/f}$ males (6 weeks) were treated with leptin (1 mg/kg body weight, ip), and hypothalami were harvested 45 min later. Tissues were homogenized in ice-cold lysis buffer (50 mM Tris HCl, pH 7.5, 0.5% Nonidet P-40, 150 mM NaCl, 2 mM EGTA, 1 mM $Na_3VO_4$, 100 mM NaF, 10 mM $Na_4P_2O_7$, 1 mM phenylmethylsulfonyl fluoride, 10 μg/ml aprotinin, 10 μg/ml leupeptin). Tissue extracts were immunoblotted with the indicated antibodies (Supplemental Table 1).

**qPCR**. qPCR was performed using Radiant™ SYBR Green 2× Lo-ROX qPCR Kits (Alkali Scientific, Pompano Beach, FL) and StepOnePlus RT PCR Systems (Life Technologies Corporation, NY, USA). Primers Ucp1-F: ATACTGGCAGATGA CGTCCC, Ucp1-R: GTACATGGACATCGCACAGC; 36B4-F: AAGCGCGTCCT GGCATTGTCT, 36B4-R: CCGCAGGGGCAGCAGTGGT.

**Electrophysiological recordings**. Mice (5 or 11 weeks) were implanted with intracerebroventricular (icv) cannula and subjected to recordings after a 1-week recover period. Mice were anesthetized with ketamine (29 mg/kg body weight, intramuscular). BAT sympathetic nerve fibers were recorded using AxoScope 10.2 (San Jose, CA). Leptin (0.5 μl/mouse, 1 mg/ml) was injected via icv cannula. Nerve activity was amplified (NL104), filtered (NL 125/126, Neurolog, Digitimer Ltd), passed 100–1000 Hz, digitized (CED 1401, Cambridge Electronic Design, Cambridge, UK), and analyzed offline using Spike2 software (Cambridge Electronic Design, Cambridge, UK). To quantify SNA, the number of action potentials crossing a pre-set threshold was determined per second. The threshold was calculated at twice the baseline nerve activity for all experiments.

**Statistical analysis**. Data were presented as means ± SEM. Differences between two groups were analyzed with two-tailed Student's $t$-test, and differences between more than two groups were analyzed using one-way and two-way analysis of variance (ANOVA) and Bonferroni posttest using GraphPad Prism 7. A $P$ value less than 0.05 was considered significant.

**Reporting summary**. Further information on research design is available in the Nature Research Reporting Summary linked to this article.

## Data availability

The authors declare that the data supporting the findings of this study are available within the paper and supplementary information files. The source data underlying Figs. 1a, 2a–d, 6d, h and 7c and Supplementary Figs 1a and 5d are provided as a Source Data file.

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

## Acknowledgements

We thank Ed Stunkel, Brian Pierchala, and Christin Carter-Su (University of Michigan) for helpful discussion and suggestions. This study was supported by grants RO1 DK094014, RO1 DK115646, DK-114220 (to L.R.), and R01 DK110436 and P30 DK34933 (to C.O.) from the National Institutes of Health and Grant #1-18-IBS-189 from the American Diabetes Association (to L.R.). This work utilized the cores supported by the Michigan Diabetes Research and Training Center (NIH DK020572), Michigan Metabolomics and Obesity Center (DK089503), and the University of Michigan Gut Peptide Research Center (NIH DK34933).

## Author contributions

L.J., H. Su, X.W., H. Shen, and M.-H.K. conducted the experiments, L.J. and L.R. designed the experiments and wrote the paper, and L.J., H. Su, X.W., H. Shen, M.-H.K., Y.L., M.G.M., C.O., and L.R. performed data analyses and edited the paper.

## Competing interests

The authors declare no competing interests.
