## [Peer Review File · Nature Communications]

Reviewers' Comments:

Reviewer #1:

Remarks to the Author:

This study examined the role of a key signaling molecule SH2B1 in the mediobasal hypothalamus in body weight through regulation of autonomic outputs to liver and fat tissues. Multiple models including Cre-mediated deletion, AAV-Cre mediated deletion and AAV-mediated overexpression, and multi-disciplinary approaches were used to demonstrate that a profound role of hypothalamic SH2B1 signaling in promoting energy expenditure through sympathetic nervous output (SNS) to fat tissues as well as increasing insulin sensitivity in liver. The experiments were well designed and the results were consistent across different animal models and experimental paradigms. Especially, the results on SNS changes in adipose tissues during ageing related to energy expenditure are novel and convincing. However, a few issues that the authors need to address to further improve the quality of the study.

1) The major concern is that the identity of hypothalamic neurons is not clear. LepR-Cre neurons represent a large number of neurons in various location and AAV-Cre injections will hit non-specifically mediobasal hypothalamic neurons. The authors at minimum should provide a colocalization profile between LepR and SH2B1, and deletion pattern of SH2B1 mediated by AAV-Cre as well as expression pattern by AAV-SH2B1. Related to this, the authors need to use a few sentences in the Discussion to acknowledge this issue.

2) SH2B1 is known to mediate leptin action. The phenotype of SH2B1 deletion in LepR neurons on reduction in energy expenditure is consistent with leptin action on energy expenditure; however, normal feeding behavior seems at odds with the leptin action on feeding. Is this due to different subsets of LepR neurons: one for feeding (using non-SH2B1) and other for energy expenditure (using SH2B1)?

3) The ageing effect on SNS in adipose tissues (normal development and gradually loss with ageing) is very interesting. Leptin signaling is known to modulate hypothalamic neuron development. This would argue that the effect on SNS is not due to disrupted leptin action, which would otherwise cause developmental effects on hypothalamic neuron and in turn cause defects in SNS innervation of adipose tissues during development. To confirm this, the authors need to examine whether deletion of SH2B1 in LepR neurons leads to hypothalamic neuron development. This result may yield the information on whether the SNS effect is regulated by leptin or other non-leptin factors.

Other issues

1) More details should be provided on methods and reagents. The AAV vector information is not complete and should include promoter and other related information.

2) The SNS recording in Fig. 6F showing leptin increased firing, which didn't reach a plateau 4 hours after injection. This is interesting and unexpected. The authors may need to clarify and discuss this with related literature with similar recording on different animal models.

Reviewer #2:

Remarks to the Author:

Sh2b1

In this investigation, the authors report a series of interesting observations related to the function of Sh2b1 (referred to in the following as S) in leptin-receptor-positive cells in the hypothalamus. At first sight, this seems to represent a coherent story, linking anti-obesity via S to increased BAT activity and increased energy expenditure.

However, there is a major problem with the paper in its present version, and that is the way the authors represent energy expenditure. Following a tradition that is being promoted in many papers – as results seem very convincing – they express energy expenditure per kg body weight. Whereas this sometimes can be reasonable, it is definitely not reasonable when animals with different

amounts of body fat are compared. This issue has been discussed over and over again, e.g. in the papers mentioned below. The point is that body fat is totally metabolically inert (i.e. not body adipose tissue but the value that come out of the pDexa that would represent chemical fat). When this fat is included in the divisor, all obese animals will necessarily demonstrate decreased energy expenditure – and the obesity is then explained. The correct way is to give this per total mouse, as indeed done here for food intake. Alternatively, it can be given per lean weight.

Although I of course only can estimate the outcome from the data presented, the differences in energy expenditure shown in Fig. 1 would largely disappear if correctly expressed (per mouse), considering the weights given for the 10-w body weights in A – and the differences in suppl fig 1 that already are very small would fully disappear. The same would be the case for the energy expenditure in Fig. 3 and Fig. 4. This has evidently wide-ranging effects for the entire paper. (This does not mean that I don't find the data publishable, even in Nat Comm but they have to be expressed in a meaningful way).

An alternative would be to express the data per lean body weight. These data are not shown but this takes us to some problem with the data for fat mass presented here. Particularly the data for females in Fig. 1 B, from 10 week old mice according to the legend: the fat content goes from 5% in f/f (should have been grey in the figure) to 40% in the deltas. I am surprised that the authors have not been confounded by this: considering the relatively small increase in total body weight at this time, it must mean that there has been a very dramatic decrease in lean body weight in these mice. This does not make sense. The authors should also present the lean mass and critically evaluate whether the values can be correct.

(Small issue: to use a 0-1 scale on RER is not meaningful for a parameter that really cannot be lower than 0.7. And the values presented are high for a chow diet. And disturbingly the values do not show the day/night variation expected, as if the mice were eating all day).

What is valid from the present fig 1 is thus mainly that S ablation decreases body temperature, a conclusion related to leptin effects that I think is fully valid.

The conclusion of the main text of the section referring to Fig. 1 is thus likely not correct: there is likely no (measurable) effect of S on energy expenditure.

Given the obesity induced according to fig 1 the data in fig 2 are fully as expected, as a consequence of the obesity. Any similar obesity would produce similar data. This thus means that I find the last sentence in the text describing fig 2 somewhat misleading: yes, S is required for "metabolic fitness" – but there is no magic here: lack of S makes the mice obese – and obesity is not good for "metabolic fitness". This should be correctly expressed.

The problems and conclusions concerning fig 3 are thus similar to what I have discussed above, including the effects of obesity as such. This would also have to be reformulated after correction of the energy expenditure to be expressed per mouse.

The same goes for fig. 4. However, I have to add here that there seem to be a calculation error in 4C: in all other panels, the y-axis runs in thousands – but here only in hundreds.

The first part of the – principally interesting – section of BAT is unfortunately difficult to evaluate. The problem is that the BAT is expanding (also according to suppl fig 1) – and this expansion largely is due to lipid accumulation (probably mainly due to the general obesity). This means that per area (or volume) everything else will be diluted. This can easily be seen on the H&E data (and correspondingly for the S overexpressors). Thus, the data on UCP1 density are probably correct but they are not meaningful. My suggestion to correct for the dilution problem would simply be to divide the UCP1 data in A with the H&E data; this would compensate for the dilution. – Let me also add already here that I am not totally happy with the TH pictures: particularly in AAV-GFP (and probably elsewhere, not good enough resolution) the staining does not look like nerves at all. Do the authors have any validation of the antibody?

Probably the most stunning effect of the S ablation is the data concerning UCP1: its total disappearance! I think this is a very important observation. – Concerning the UCP1 mRNA values I

would like to see them directly expressed as UCP1/36B4 (not normalized to 1) so that the level can be directly compared to the level in inguinal adipose tissue in the suppl fig. – The experiments in 5D are also such that are transmitted from investigation to investigation despite conceptual criticism: keeping body temperature when suddenly exposed to 4 °C while coming from normal animal house temperatures (not specified in the present paper) is mainly accomplished via shivering thermogenesis; acute cold tolerance probably does not reflect BAT capacity. Still, the outcome is the outcome – but whether it is related to the presence of UCP1 cannot be deduced from such experiments.

The increase in leptin in fig 6 is referred to in the text as a sign of leptin resistance. This is hardly the case: the leptin levels are likely simply showing the increase in body fat. In DE it is confusing that the traces shown in D clearly indicate a higher baseline in the deltas than in the f/f – while the compilation in E shows the opposite. However, the data on total absence of leptin effect are of course very convincing. The implications of S ablation for the mediation of cold stimulus in GH look very interesting but have perhaps not been fully explored.

The whole observation in fig 7 seems also very interesting (I suppose the authors mean 8 weeks on first line page 9). Unfortunately, the DAPI color does not transmit well so that cell density is not easy to follow; could the authors not change DAPI color to something visible?

All in all, I find many of the observations presented here of general interest. The effects on maintained body temperature (that is not a hypothermia as the authors write bottom page 6; a hypothermia is what is shown in fig 5D: an inability to produce sufficient heat giving high cooling; this is not related to the change in body temperature control so convincingly shown in suppl fig 1 C where the mice consequently regulated body temperature 1 °C below) are in good extension from what is known on leptin function but now more detailed analysed in mediation, and the effects on UCP1 gene expression and mediation of cold stimulus and the effect of aging are all of interest. The energy expenditure part is as indicated probably not correctly analysed – and the effects of induced obesity are what they always are. I would suggest that the authors rewrite the outcome and have it re-evaluated for the journal.

References

Himms-Hagen, J. (1997). On raising energy expenditure in ob/ob mice. *Science* 276, 1132.
Butler, A. A. and Kozak, L. P. (2010). A recurring problem with the analysis of energy expenditure in genetic models expressing lean and obese phenotypes. *Diabetes* 59, 323-329.

Reviewer #3:

Remarks to the Author:

This paper reports effects of gain or loss of function of Sh2b1 in the mediobasal hypothalamus on body weight, energy expenditure and a lesser effect on food intake. Studies using LepR are mice suggest that most of these effects are mediated by neurons expressing the leptin receptor.

However the authors should address the following comments :

It is unclear which specific neural populations express Sh2b1 and it is possible that some of the effects that are observed might be mediated by one or more of these other subpopulations. It would be useful to know if Sh2b1 is expressed in POMC, AGRP or other neurons in the MBH that have been reported to modulate body weight. This is important in light of the fact that leptin has contrasting effects on these ARC neurons, (ie Agrp and POMC), and since the MBH deletion of a component of the JAK signaling pathway, it is unclear which effects might be the dominant ones. Given the existence of numerous cre lines targeting specific MBH LepR populations, in Fig. 3 it is important to distinguish between each of these populations.

The authors write: "Unexpectedly, food intake was increased by MBH-specific ablation of Sh2b1 (by AAV-cre injections) (Fig. 3E), suggesting that Sh2b1 in MBH non-LepR neurons suppresses

food intake." This augments the importance of defining the cell types which express this gene. In addition one could test this directly by delivering AAVs in which cre inactivates Sh2b1 into the LepR-cre X Sh2b1 fl/fl animals.

With respect to Fig. 4 (overexpression of Sh2b1 in the MBH), here again the specific neuron population expression of Sh2b1 should be explored (ie Agrp/pomc etc). Furthermore, it is surprising (based on the western blot shown here) that there is little apparent Sh2b1 expression in the MBH, ie what is the baseline expression levels of Sh2b1, and how many fold is it increasing it using AAV.

In Supp Fig. 2 for targeting of AAV vectors: 1) DAPI should be shown, to validate actual coordinates of injection/spread of virus. 2) Targeting of AAV for overexpression of Sh2b1 in the hypothalamus should be shown as well, not shown just by western blotting, given that no cre lines are used here for specificity.

The data in Fig. 7 do not support the assertion that sh2b1 is a critical regulator of BAT SNS. If BAT SNS is normal at 8 weeks, but deteriorates over time, this instead suggests that sh2b1 could be involved in the maintenance of BAT SNS although these observations could also be a result of the obesity itself. In fact it could also be that the levels of TH have been "diluted" because of the weight change between 8-22 weeks (20->40g KO vs 20->28g in f/f), or that TH expression might have decreased while the number of neurons stays the same.

The level of TH staining appears different across the different fields of view and some of staining appears to be background vs fibers in Figs. 5 and 7 as well. The colors in the futures are also different (magenta vs red) and it would be helpful if the colors were consistent.

At the bottom of page 6 the authors state "Given the hypothermia phenotypes of Sh2b1 Δ LepR mice". However data supporting this isn't shown initially nor is the literature cited for this. This phrase should be at the end of the paragraph not at the beginning given the evidence is only shown at the end. What is the effect on temperature in the BAT when these mice are cold challenged. Core body temperature falling faster (increased hypothermia) could occur for numerous reasons that are not BAT related (ie changes in vasodilation, shivering).

The language is unclear and in some cases this make it difficult to evaluate the paper. The paper should be carefully edited to correct the numerous grammatical/language errors and improve the writing.

Response to Reviewers' comments

Response to Reviewer #1:

“This study examined the role of a key signaling molecule SH2B1 in the mediobasal hypothalamus in body weight through regulation of autonomic outputs to liver and fat tissues. Multiple models including Cre-mediated deletion, AAV-Cre mediated deletion and AAV-mediated overexpression, and multi-disciplinary approaches were used to demonstrate that a profound role of hypothalamic SH2B1 signaling in promoting energy expenditure through sympathetic nervous output (SNS) to fat tissues as well as increasing insulin sensitivity in liver. The experiments were well designed and the results were consistent across different animal models and experimental paradigms. Especially, the results on SNS changes in adipose tissues during ageing related to energy expenditure are novel and convincing. However, a few issues that the authors need to address to further improve the quality of the study”.

1) “The major concern is that the identity of hypothalamic neurons is not clear. LepR-Cre neurons represent a large number of neurons in various location and AAV-Cre injections will hit non-specifically mediobasal hypothalamic neurons. The authors at minimum should provide a colocalization profile between LepR and SH2B1, and deletion pattern of SH2B1 mediated by AAV-Cre as well as expression pattern by AAV-SH2B1. Related to this, the authors need to use a few sentences in the Discussion to acknowledge this issue”.

We thank Reviewer 1 for these important comments. We performed new experiments and added new data, following these comments. Using both anti-Sh2b1 immunostaining and newly-generated Sh2b1-Cre;Rosa-mTmG reporter mice, we showed that Sh2b1 is ubiquitously expressed in the entire hypothalamus, including POMC and AgRP neurons (new revised Supplementary Fig. 4A-C). Because commercial anti-LepR antibodies are not suitable for immunostaining assays, we are unable to examine colocalization of Sh2b1 and LepR. We provided new results showing the patterns of AAV-Cre-mediated deletion (revised Supplementary Fig. 2A) and AAV-SH2B1-induced expression (revised Supplementary Fig. 3A). To further address hypothalamic subpopulations, we generated and characterized new POMC neuron-specific Sh2b1 knockout mice. Unlike LepR neuron-specific knockout, POMC neuron-specific Sh2b1 knockout did not alter body weight and metabolism (revised Supplementary Fig. 5A-E). We postulate that Sh2b1 pathways in the other subpopulations may functionally compensate for the loss of Sh2b1 in POMC neurons. In agreement with this notion, ablation of LepR in individual subpopulations (e.g. POMC, AgRP, Ghrh, Htr2c, Prlh, Sf1, Sim1), unlike db/db mice, has been reported to minimally (or not) affect body weight and metabolism. We expanded Discussion and acknowledged Sh2b1 neuron populations as suggested. We wish to point out that identification LepR neurons and MBH neurons as Sh2b1 targets advance, in our view, our understanding of the regulation of body weight and metabolism by Sh2b1 and leptin.

2) “SH2B1 is known to mediate leptin action. The phenotype of SH2B1 deletion in LepR neurons on reduction in energy expenditure is consistent with leptin action on energy expenditure; however, normal feeding behavior seems at odds with the leptin action on feeding.

Is this due to different subsets of LepR neurons: one for feeding (using non-SH2B1) and other for energy expenditure (using SH2B1)”?

Following these comments, we expanded discussion about feeding vs energy expenditure. As Reviewer 1 pointed out, the Sh2b1 and non-Sh2b1 populations of LepR neurons may regulate energy expenditure and feeding, respectively. Alternatively, the Sh2b1 and non-Sh2b1 branches of leptin signaling pathways in the same or overlapping LepR neurons may regulate energy expenditure and feeding, respectively. We will test this notion in the future (beyond the scope of this study in our view).

3) “The ageing effect on SNS in adipose tissues (normal development and gradually loss with ageing) is very interesting. Leptin signaling is known to modulate hypothalamic neuron development. This would argue that the effect on SNS is not due to disrupted leptin action, which would otherwise cause developmental effects on hypothalamic neuron and in turn cause defects in SNS innervation of adipose tissues during development. To confirm this, the authors need to examine whether deletion of SH2B1 in LepR neurons leads to hypothalamic neuron development. This result may yield the information on whether the SNS effect is regulated by leptin or other non-leptin factors”.

Following these comments, we emphasized that Sh2b1 in LepR neurons critically supports the maintenance but not development of the SNS. We agree that examination of the effect of Sh2b1 on hypothalamic development likely leads to new discoveries. We wish to point out that the hypothalamic development study requires additional models and reagents (currently unavailable to us) and is time-consuming. We will study this important question in the future (beyond the scope of this study).

Other issues

1) More details should be provided on methods and reagents. The AAV vector information is not complete and should include promoter and other related information.

We have provided the requested AAV vector information in the Method-Stereotaxic microinjection. Cre expression is driven by the human Synapsin promoter; The SH2B1 is under the control of the constitutively active synthetic CAG promoter.

2) The SNS recording in Fig. 6F showing leptin increased firing, which didn't reach a plateau 4 hours after injection. This is interesting and unexpected. The authors may need to clarify and discuss this with related literature with similar recording on different animal models.

Following these comments, we expanded discussion (e.g. multiple synaptic modifications and polysynaptic connection). We cited additional literature showing a similar pattern of the leptin action on BAT SNS (JCI, 100:270-278, 1997).

Response to Reviewer #2:

“In this investigation, the authors report a series of interesting observations related to the function of Sh2b1 (referred to in the following as S) in leptin-receptor-positive cells in the hypothalamus. At first sight, this seems to represent a coherent story, linking anti-obesity via S to increased BAT activity and increased energy expenditure. However, there is a major problem with the paper in its present version, and that is the way the authors represent energy expenditure. Following a tradition that is being promoted in many papers – as results seem very convincing – they express energy expenditure per kg body weight. Whereas this sometimes can be reasonable, it is definitely not reasonable when animals with different amounts of body fat are compared. This issue has been discussed over and over again, e.g. in the papers mentioned below. The point is that body fat is totally metabolically inert (i.e. not body adipose tissue but the value that come out of the pDexa that would represent chemical fat). When this fat is included in the divisor, all obese animals will necessarily demonstrate decreased energy expenditure – and the obesity is then explained. The correct way is to give this per total mouse, as indeed done here for food intake. Alternatively, it can be given per lean weight. Although I of course only can estimate the outcome from the data presented, the differences in energy expenditure shown in Fig. 1 would largely disappear if correctly expressed (per mouse), considering the weights given for the 10-w body weights in A – and the differences in suppl fig 1 that already are very small would fully disappear. The same would be the case for the energy expenditure in Fig. 3 and Fig. 4. This has evidently wide-ranging effects for the entire paper. *(This does not mean that I don't find the data publishable, even in Nat Comm but they have to be expressed in a meaningful way)*. An alternative would be to express the data per lean body weight”.

We thank Reviewer 2 for these important comments, and have recalculated energy expenditure (normalized to mice) as requested. We added the new calculations in the revised Supplementary Fig. 1D-E, Supplementary Fig. 2D and Supplementary Fig. 3C. Given that there is currently no consensus on data normalization in the field (e.g. normalizations to mice, lean mass, body weight, or metabolic body weight), we also kept the original data; however, we deleted the data interpretation of Sh2b1 regulation of energy expenditure. In the revised Discussion, we added statements: “It is worth mentioning that interpretation of global energy expenditure data was confounded by normalization. Energy expenditure values were lower in Sh2b1^{ΔLepR} mice when normalized to body weight, but was normal upon normalization to mice. We also did not observe the RER circadian likely due to assay-related stress. Thus, the impact of neuronal Sh2b1 on global energy expenditure needs to be further analyzed in the future”.

“These data are not shown but this takes us to some problem with the data for fat mass presented here. Particularly the data for females in Fig. 1 B, from 10 week old mice according to the legend: the fat content goes from 5% in f/f (should have been grey in the figure) to 40% in the deltas. I am surprised that the authors have not been confounded by this: considering the relatively small increase in total body weight at this time, it must mean that there has been a very dramatic decrease in lean body weight in these mice. This does not make sense. The authors should also present the lean mass and critically evaluate whether the values can be correct. (Small issue: to use a 0-1 scale on RER is not meaningful for a parameter that really

cannot be lower than 0.7. And the values presented are high for a chow diet. And disturbingly the values do not show the day/night variation expected, as if the mice were eating all day”.

We rechecked the original data and pDexa settings, and found that the underestimation of fat content was caused by pDexa settings. We replaced the female fat content data with new results from female mice at 20 weeks of age (the revised Fig. 1B). We also added lean mass as suggested (the revised Supplementary Fig. 1B). Of Note, heads, necks and tails were unable to be included in imaging analysis by the pDexa (added to the revised methods), so the calculated values of fat content and lean mass are under-estimated. Slightly-high values of RER may be caused by metabolic cage-intrinsic physical properties (e.g. settings and O₂ and CO₂ electrodes). Lack of RER circadian changes may be caused by stress during assays. We acknowledged these caveats in the revised manuscript.

“What is valid from the present fig 1 is thus mainly that S ablation decreases body temperature, a conclusion related to leptin effects that I think is fully valid. The conclusion of the main text of the section referring to Fig. 1 is thus likely not correct: there is likely no (measurable) effect of S on energy expenditure”.

Following these comments, we deleted the statements about the relationship of Sh2b1 with global energy expenditure, and kept the conclusions about the impact of Sh2b1 on body temperature.

“Given the obesity induced according to fig 1 the data in fig 2 are fully as expected, as a consequence of the obesity. Any similar obesity would produce similar data. This thus means that I find the last sentence in the text describing fig 2 somewhat misleading: yes, S is required for “metabolic fitness” – but there is no magic here: lack of S makes the mice obese – and obesity is not good for “metabolic fitness”. This should be correctly expressed”.

We deleted “metabolic fitness” and revised the expression, following these comments. It is worth mentioning that obesity is not always associated with metabolic disease (so called healthy obesity”. Metabolically-healthy obesity has been repeatedly described in literature. We would like to show that Sh2b1 deficiency is not related to “healthy obesity”.

“The problems and conclusions concerning fig 3 are thus similar to what I have discussed above, including the effects of obesity as such. This would also have to be reformulated after correction of the energy expenditure to be expressed per mouse”.

Following these comments, we reassessed energy expenditure (normalized to mice) (the revised Supplementary Fig. 2D). We deleted the statements about the relationship of Sh2b1 with global energy expenditure.

“The same goes for fig. 4. However, I have to add here that there seem to be a calculation error in 4C: in all other panels, the y-axis runs in thousands – but here only in hundreds”.

We reassessed energy expenditure by normalization to mice (the revised Supplementary Fig. 3C). We made correction on the revised Fig. 4C.

“The first part of the – principally interesting – section of BAT is unfortunately difficult to evaluate. The problem is that the BAT is expanding (also according to suppl fig 1) – and this expansion largely is due to lipid accumulation (probably mainly due to the general obesity). This means that per area (or volume) everything else will be diluted. This can easily be seen on the H&E data (and correspondingly for the S overexpressors). Thus, the data on UCP1 density are probably correct but they are not meaningful. My suggestion to correct for the dilution problem would simply be to divide the UCP1 data in A with the H&E data; this would compensate for the dilution. – Let me also add already here that I am not totally happy with the TH pictures: particularly in AAV-GFP (and probably elsewhere, not good enough resolution) the staining does not look like nerves at all. Do the authors have any validation of the antibody?”

We normalized Ucp1 density to H&E areas as suggested (Figure 1 for reviewer 2), and the results support the original conclusions. However, normalization to neither BAT (i.e. index for cell size; normalization to cell size/volume) nor H&E areas appears to be perfect in our view. Therefore, we deleted the quantification results and just showed representative images.

We previously validated the anti-TH antibody (JCI, 130:2305-2317, 2019). Sympathetic denervation completely eliminated immunoreactivity to TH in BAT (Figure 2 for Reviewer 2 and Reviewer 3 [Redacted]). We cited anti-TH antibody validation and this paper in the revised Results. With regard to the TH image morphology and size, BAT sections were cut across nerve fibers (i.e. lines), cross-sections of fibers (small dots) or axonal varicosities (large dots). Neuron cell bodies are located in the sympathetic ganglia outside of BAT. We wish to point out that it is technically challenging to quantify the number of BAT sympathetic neurons because they are mingled with other neurons projecting to the lung, heart, and other internal organs.

“Probably the most stunning effect of the S ablation is the data concerning UCP1: its total disappearance! I think this is a very important observation. – Concerning the UCP1 mRNA values I would like to see them directly expressed as UCP1/36B4 (not normalized to 1) so that the level can be directly compared to the level in inguinal adipose tissue in the suppl fig. – The experiments in 5D are also such that are transmitted from investigation to investigation despite conceptual criticism: keeping body temperature when suddenly exposed to 4 °C while coming from normal

Figure 1 for Reviewer 2.

[Redacted]

animal house temperatures (not specified in the present paper) is mainly accomplished via shivering thermogenesis; acute cold tolerance probably does not reflect BAT capacity. Still, the outcome is the outcome – but whether it is related to the presence of UCP1 cannot be deduced from such experiments”.

We normalized Ucp1 expression to 36B4 levels as required (the revised Fig. 5C, the revised Supplementary Fig. 1G). We wish to point out that in literature, deletion of Ucp1 or other key genes responsible for BAT thermogenesis leads to acute cold intolerance. Nonetheless, we removed the statements which might suggest that Ucp1 deficiency alone is responsible for acute cold intolerance, following these comments.

“The increase in leptin in fig 6 is referred to in the text as a sign of leptin resistance. This is hardly the case: the leptin levels are likely simply showing the increase in body fat. In DE it is confusing that the traces shown in D clearly indicate a higher baseline in the deltas than in the f/f – while the compilation in E shows the opposite. However, the data on total absence of leptin effect are of course very convincing. The implications of S ablation for the mediation of cold stimulus in GH look very interesting but have perhaps not been fully explored”.

We deleted the statements about potential relationships between hyperleptinemia and leptin resistance, following these comments. With regard to DE, it is not uncommon in electrophysiological recordings that background noise signals (i.e. trace thickness) slightly vary. The spikes represent action potential and neuronal activation. Nonetheless, we replaced recording traces, following these comments (the revised Fig. 6D). Both baseline and leptin-stimulated SNA (spikes) were lower in Sh2b1^{ALepR} mice. These results provide proof of concept evidence defining the essential role of Sh2b1. An independent study is warranted to delineate in depth the molecular and cellular mechanisms of the Sh2b1 action (beyond the scope of this work in our view).

“The whole observation in fig 7 seems also very interesting (I suppose the authors mean 8 weeks on first line page 9). Unfortunately, the DAPI color does not transmit well so that cell density is not easy to follow; could the authors not change DAPI color to something visible”?

It was 6 on page 9. We increased DAPI intensity as suggested (the revised Fig. 7A).

“All in all, I find many of the observations presented here of general interest. The effects on maintained body temperature (that is not a hypothermia as the authors write bottom page 6; a hypothermia is what is shown in fig 5D: an inability to produce sufficient heat giving high cooling; this is not related to the change in body temperature control so convincingly shown in suppl fig 1 C where the mice consequently regulated body temperature 1 °C below) are in good extension from what is known on leptin function but now more detailed analysed in mediation, and the effects on UCP1 gene expression and mediation of cold stimulus and the effect of aging are all of interest. The energy expenditure part is as indicated probably not correctly analysed – and the effects of induced obesity are what they always are. I would suggest that the authors rewrite the outcome and have it re-evaluated for the journal”.

We deleted “hypothermia”, following these comments. We re-assessed energy expenditure, reanalyzed the results, and rewrote the manuscript, following these comments.

Response to Reviewer #3:

“This paper reports effects of gain or loss of function of Sh2b1 in the mediobasal hypothalamus on body weight, energy expenditure and a lesser effect on food intake. Studies using LepR are mice suggest that most of these effects are mediated by neurons expressing the leptin receptor. However the authors should address the following comments:

It is unclear which specific neural populations express Sh2b1 and it is possible that some of the effects that are observed might be mediated by one or more of these other subpopulations. It would be useful to know if Sh2b1 is expressed in POMC, AGRP or other neurons in the MBH that have been reported to modulate body weight. This is important in light of the fact that leptin has contrasting effects on these ARC neurons, (ie Agrp and POMC), and since the MBH deletion of a component of the JAK signaling pathway, it is unclear which effects might be the dominant ones. Given the existence of numerous cre lines targeting specific MBH LepR populations, in Fig. 3 it is important to distinguish between each of these populations.

The authors write: “Unexpectedly, food intake was increased by MBH-specific ablation of Sh2b1 (by AAV-cre injections) (Fig. 3E), suggesting that Sh2b1 in MBH non-LepR neurons suppresses food intake.” This augments the importance of defining the cell types which express this gene. In addition one could test this directly by delivering AAVs in which cre inactivates Sh2b1 into the LepR-cre X Sh2b1 fl/fl animals”.

We thank Reviewer 3 for these important comments. Following these advices, we attempted to further map and characterize specific neuron populations expressing Sh2b1. Immunostaining with anti-Sh2b1 antibody showed that most of hypothalamic neurons express Sh2b1 (the revised Supplementary Fig. 4A). To confirm these results, we generated Sh2b1-Cre knockin mice, and crossed Sh2b1-Cre mice with Rosa-mTmG reporter mice to genetically label Sh2b1-expressing cells with GFP. Consistently, most of hypothalamic cells were GFP-labelled (the revised Supplementary Fig. 4B). To test if POMC and/or AgRP neurons express Sh2b1, we prepared brain sections from Sh2b1-Cre;Rosa-mTmG reporter mice (GFP as a surrogate marker for Sh2b1) and performed immunostaining with anti-POMC and anti-AgRP antibodies. Both POMC and AgRP neurons expressed Sh2b1 (the revised Supplementary Fig. 4C). We are unable to perform coimmunostaining of Sh2b1 with POMC or AgRP, because anti-Sh2b1, anti-POMC, and Anti-AgRP antibodies all were raised from rabbits.

Following these comments, we further generated and characterized POMC neuron-specific Sh2b1 knockout mice. Unlike LepR neuron-specific knockout, POMC neuron-specific Sh2b1 knockout did not alter body weight and metabolism (the revised Supplementary Fig. 5A-E). We postulate that Sh2b1 pathways in the other subpopulations may functionally compensate for the loss of Sh2b1 in POMC neurons. In agreement with this notion, ablation of LepR in individual subpopulations (e.g. POMC, AgRP, Ghrh, Htr2c, Prlh, Sf1, Sim1), unlike db/db mice, only slightly (or not) affects body weight and metabolism. In the future, we will generate and characterize additional cell type-specific Sh2b1 knockout mice to further test hypothalamic Sh2b1. We wish to highlight that identification of LepR and MBH neurons as Sh2b1 targets importantly advances our understanding of the regulation of body weight and metabolism by Sh2b1 and leptin.

Sh2b1^{ALepR} mice are normal in food take, whereas MBH-specific Sh2b1 knockout mice are hyperphagic. These results clearly demonstrate that Sh2b1 in non-LepR neurons

suppresses feeding. The suggested AAV-Cre injection into LepR-cre X Sh2b1^{fl/fl} mice experiments are expected to provide additional data to further confirm these results, and they are unlikely to generate new information beyond what described in the manuscript. We wish to point out that it will take a long time and a large effort to raise enough LepR-cre X Sh2b1^{fl/fl} mice for these experiments. Hence, we will delay the experiments and perform them in the future in a separate study aiming to elucidate Sh2b1-related feeding circuits (this work is focused on the energy expenditure pathway).

“With respect to Fig. 4 (overexpression of Sh2b1 in the MBH), here again the specific neuron population expression of Sh2b1 should be explored (ie Agrp/pomc etc). Furthermore, it is surprising (based on the western blot shown here) that there is little apparent Sh2b1 expression in the MBH, ie what is the baseline expression levels of Sh2b1, and how many fold is it increasing it using AAV”.

We wish to point out that it will take a long time and an enormous amount of efforts to generate mice with neuron type-specific overexpression of Sh2b1. Based on the findings from neuron type-specific LepR knockout mice and our new data from POMC neuron-specific Sh2b1 knockout mice (the revised Supplementary Fig. 5A-E), Sh2b1 and LepR signaling in different populations may have similar functions. It is unlikely to examine and quantify the contribution of each individual Sh2b1 subpopulation to the metabolic phenotypes in one paper.

We replaced the Sh2b1 western blots showing endogenous Sh2b1 in the MBH (the revised Supplementary Fig. 3A). The levels of endogenous Sh2b1 are underestimated because of short exposure time (due to super-physiological levels of overexpressed SH2B1). We acknowledged the confounding effect of super-physiological SH2B1 on data interpretation. However, our conclusions were based on both deletion of endogenous Sh2b1 and overexpression of SH2B1 (not just on overexpression alone). We are unable to perform co-staining experiments (i.e. Sh2b1 with POMC or AgRP) because antibodies to Sh2b1, POMC, and AgRP all were raised in rabbits.

“In Supp Fig. 2 for targeting of AAV vectors: 1) DAPI should be shown, to validate actual coordinates of injection/spread of virus. 2) Targeting of AAV for overexpression of Sh2b1 in the hypothalamus should be shown as well, not shown just by western blotting, given that no cre lines are used here for specificity”.

We added the requested results in the revised Supplementary Fig. 2A and Fig. 3A.

“The data in Fig. 7 do not support the assertion that sh2b1 is a critical regulator of BAT SNS. If BAT SNS is normal at 8 weeks, but deteriorates over time, this instead suggests that sh2b1 could be involved in the maintenance of BAT SNS although these observations could also be a *result of the obesity itself*. In fact it could also be that the levels of TH have been “diluted” because of the weight change between 8-22 weeks (20->40g KO vs 20->28g in f/f), or that TH expression might have decreased while the number of neurons stays the same. The level of TH staining appears different across the different fields of view and some of staining appears to be background vs fibers in Figs. 5 and 7 as well. The colors in the futures are also different (magenta vs red) and it would be helpful if the colors were consistent”.

Following these comments, we further clarified the role of Sh2b1 in the maintenance but not development of the SNS/BAT axis. We also acknowledged the potential influence of obesity on the SNS/BAT axis. Reviewer 1 and Reviewer 2 commented on the data in Fig. 7 “very interesting”. To distinguish between TH expression and nerve fiber deterioration, we stained BAT section with TUJ1, a distinct SNS nerve marker. TUJ1 levels, like TH signals, were also markedly reduced in Sh2b1^{ΔLepR} mice (the revised Fig. 7C). In accordance with these results, both baseline and leptin-stimulated BAT sympathetic nerve activities were markedly lower in Sh2b1^{ΔLepR} mice (Fig. 6E-F). To address the concern about fat dilution, we use western blots to examine phosphorylation of HSL (stimulated by the SNS), and normalized HSL phosphorylation adipose protein levels (reducing fat dilution). HSL phosphorylation became lower in Sh2b1^{ΔLepR} mice in an age-dependent manner (Fig. 7E). Collectively, these data indicate that the BAT SNS deteriorates progressively in Sh2b1^{ΔLepR} mice.

We previously validated the anti-TH antibody (JCI, 130:2305-2317, 2019). Sympathetic denervation completely eliminated immunoreactivity to TH in BAT (Figure 2 for Reviewer 2 and Reviewer 3). We cited this paper and anti-TH antibody validation in the revised Results. Section locations influence the size and morphology of TH images. BAT sections were cut through nerve fibers (i.e. TH lines), cross-sections of fibers (small dots), or axonal varicosities (large dots). TH neuron cell bodies are located in sympathetic ganglia outside of BAT. We wish to point out that it is technically challenging to quantify the number of BAT sympathetic neurons because they are mingled with other neurons projecting to the lung, heart, and other internal organs. With the concern about color, the images were a merge of TH (red) and DAPI (blue). Magenta color indicates overlaps of TH nerves (red) with brown adipocyte nuclei (blue).

“At the bottom of page 6 the authors state “Given the hypothermia phenotypes of Sh2b1ΔLepR mice”. However data supporting this isn’t shown initially nor is the literature cited for this. This phrase should be at the end of the paragraph not at the beginning given the evidence is only shown at the end. What is the effect on temperature in the BAT when these mice are cold challenged. Core body temperature falling faster (increased hypothermia) could occur for numerous reasons that are not BAT related (ie changes in vasodilation, shivering)”.

Following these comments, we deleted “Given the hypothermia....” and rewrote the paper.

“The language is unclear and in some cases this make it difficult to evaluate the paper. The paper should be carefully edited to correct the numerous grammatical/language errors and improve the writing”.

We extensively edited the manuscript to minimize language errors.

Reviewers' Comments:

Reviewer #1:

Remarks to the Author:

The authors have provided new data with new animal lines Sh2b1-Cre, which have largely addressed my concerns on neuron identity that were manipulated in this study. Further clarifications have also been provided on developmental versus maintaining effects of Leptin-Sh2b1 signaling on SNS. Given the interesting effect of ageing on SNS that can be regulated by Leptin-sh2b1, I think this study would be interesting to broad readership of Nature Communications. I have no further concerns.

After careful review of comments from Reviewer 3 and the authors' responses. I think the authors largely addressed the concerns raised by reviewer 3; however there are some additional concerns remaining.

- 1) The authors nicely demonstrated the effect of Sh2B1 knockout specifically in POMC neurons. However, no additional data on AgRP neurons were provided. In my opinion, lacking data from AgRP neurons does not affect the conclusion that Sh2B1 in LepR neurons play a novel role in maintaining SNS tone to adipose tissues, and thus shouldn't represent a major deficiency for this manuscript.
- 2) The authors' response on SNS changes in KO mice further strengthened the story.
- 3) Although the authors provided a new sh2b1-Cre mouse strain, the data from this line were not in high quality. It is unknown why the authors chose to use Rosa-mTmG reporter strain, in which the reporters (Tomato and GFP) are membrane bound and will thus not label neurons clearly. Thus, the data quality of colocalization between Sh2b1 and POMC/AgRP is poor. In my opinion, this set of colocalization data needs to be improved.
- 4) The authors failed to provide convincing anatomical data to demonstrate successful delivery of AAV-Cre and AAV-Sh2B1 to bilateral mediobasal hypothalamus. These sets of data are important for readers to appreciate which brain regions are implicated. The readers may also wonder the expression of the virus as these viruses are not Cre-dependent and may diffuse. The authors did provide AAV-GFP injection in the supplementary data, but only with one side.
- 5) All GTTs and ITTs were performed in mice with a large difference in body weight, and the observed differences in these tests may be secondary to obesity. Thus, these sets of data should receive less emphasis and are suggested to be placed in the supplementary category.

Reviewer #2:

Remarks to the Author:

The authors have come a long way in improving the paper. With a little effort more, I believe we can arrive at an interesting and well-analyzed paper.

1. The authors have now recalculated energy expenditure data and have arrived at the expected result: that there is no difference between the groups. Although the authors in the text take full consequence of this, they have not taken the consequence in the figs. Thus, the data in suppl fig 1 DE must be moved into the main text. I personally do not think that the data per kg body weight should be shown at all but if the authors insist, they must be moved into the suppl.
2. The authors are not systematic on the "coloring" of the data. They consistently show deltaLepR as black but they jump between grey and white for f/f and LepR-cre. Not very important but there is no reason not to be systematic in this respect.
3. The authors show fat and lean masses as %, likely giving misleading effects of apparent loss of lean mass (fig S1B). The authors should show the data in g.
4. Fig S1F is one of the most interesting outcomes of the study and it should definitely be upgraded to be included in the main figs.
5. The word homeostasis in line 90 is perhaps not really optimal. The mice seem at least concerning body temperature to be homeostatic, although at a lower body temp than normal mice.

6. Evidently, the same concerning showing the data per mouse and not per kg bw is relevant for fig 3 c (to be replaced by fig S2D). Similarly the fat content in fig. 3B and S2C should be in g not %.
7. Line 124: energy expenditure is not regulated according to these data
8. I wondered whether the stress that the authors suggest explains the lack of shift in RER also have masked any real differences in energy expenditure? The authors could mention this in the Discussion.
9. Same comments again for showing the total energy expenditure (not per kg bw) and the fat content in g in the main figures, for fig 4.
10. Interesting that body temp is increased (fig 4 D). Again, these body temp data are underplayed by the authors.
11. Blood glucose in fig 3 and 4 insulin tolerance should be given as mg/ml, not %.
12. Body temp is regulated by itself and is normally not secondary to BAT activity. E.g. when normal mice are transferred to cold (as in fig 5D) they mainly keep their body temp up by shivering. Thus, in my opinion, the change in body temperature setting observed here is a direct effect of the leptin system, not secondary to BAT activity. Thus, the "accordingly" line 155 is not correct.
13. Line 174: shouldn't this be fig. 6E?
14. (Comment omitted)
15. I am surprised concerning the UCP1 mRNA data in 5c (supposedly BAT) versus 1g (stated as iWAT). Normally when expressed versus house-keeping genes, UCP1 mRNA levels are 10-100-fold lower in iWAT than in BAT. Could the authors check their calculations?
16. I was first surprised that the "controls" in 5D left and right were different – but then this may be due to the right ones being obese. This may be pointed out.
17. Of course the confusing sentences around line 250 are avoided when the unnormalized figs are shown instead.
18. Line 277: do the authors mean "only slightly"?
19. It is strange that the authors do not mention the very consistent and interesting data on body temperature regulation at all in the discussion. The study is in my opinion a very clear demonstration of the significance of leptin as a body temperature regulator. Perhaps this should also be included in the title.
20. The authors several times write as if they have measured BAT thermogenesis, including in the abstract. This they have not done; they have "only" shown molecular evidence implying an attenuation of thermogenesis.
21. Line 28 (abstract): this about energy expenditure is a little doubtful, considering what is discussed above.

With the above updates I feel that the paper will be a very interesting contribution.

Response to Reviewer 1

“The authors have provided new data with new animal lines Sh2b1-Cre, which have largely addressed my concerns on neuron identity that were manipulated in this study. Further clarifications have also been provided on developmental versus maintaining effects of Leptin-Sh2b1 signaling on SNS. Given the interesting effect of ageing on SNS that can be regulated by Leptin-sh2b1, I think this study would be interesting to broad readership of Nature Communications. I have no further concerns”.

We greatly appreciate insightful comments. Thanks.

Response to Reviewers 1/3

“After careful review of comments from Reviewer 3 and the authors' responses. I think the authors largely addressed the concerns raised by reviewer 3; however there are some additional concerns remaining”.

1) “The authors nicely demonstrated the effect of Sh2B1 knockout specifically in POMC neurons. However, no additional data on AgRP neurons were provided. In my opinion, lacking data from AgRP neurons does not affect the conclusion that Sh2B1 in LepR neurons play a novel role in maintaining SNS tone to adipose tissues, and thus shouldn't represent a major deficiency for this manuscript”.

We agree with and appreciate these assessments.

2) “The authors' response on SNS changes in KO mice further strengthened the story”.

We appreciate this comment.

3) “Although the authors provided a new sh2b1-Cre mouse strain, the data from this line were not in high quality. It is unknown why the authors chose to use Rosa-mTmG reporter strain, in which the reporters (Tomato and GFP) are membrane bound and will thus not label neurons clearly. Thus, the data quality of colocalization between Sh2b1 and POMC/AgRP is poor. In my opinion, this set of colocalization data needs to be improved”.

We performed new experiments and replaced the original Fig. S4 with new images, following these comments. These new results confirmed that Sh2b1 neurons in the ARC express POMC.

4) “The authors failed to provide convincing anatomical data to demonstrate successful delivery of AAV-Cre and AAV-Sh2B1 to bilateral mediobasal hypothalamus. These sets of data are important for readers to appreciate which brain regions are implicated. The readers may also wonder the expression of the virus as these viruses are not Cre-dependent and may diffuse. The authors did provide AAV-GFP injection in the supplementary data, but only with one side”.

We repeated AAV-GFP injection experiments and replaced the original Figs. S2A with new images, following these comments. The new results confirmed that AAV-GFP vectors were bilaterally injected into targeted hypothalamic areas. We did not repeat AAV-Sh2b1 injection site experiments, because it was injected into similar areas using the same brain coordinates.

5) “All GTTs and ITTs were performed in mice with a large difference in body weight, and the observed differences in these tests may be secondary to obesity. Thus, these sets of data should receive less emphasis and are suggested to be placed in the supplementary category”.

We would like to point out, respectfully, that obesity is not always associated with insulin resistance. Metabolically-healthy obesity (i.e. with normal insulin sensitivity and glucose metabolism) has been reported in both rodents and humans. It is important to show that LepR neuron-specific ablation of Sh2b1 causes both obesity and metabolic disorders, in our view. Therefore, the GTT and ITT data were maintained in main figures.

Response to Reviewer 2

“The authors have come a long way in improving the paper. With a little effort more, I believe we can arrive at an interesting and well-analyzed paper”.

1. “The authors have now recalculated energy expenditure data and have arrived at the expected result: that there is no difference between the groups. Although the authors in the text take full consequence of this, they have not taken the consequence in the figs. Thus, the data in suppl fig 1 DE must be moved into the main text. I personally do not think that the data per kg body weight should be shown at all but if the authors insist, they must be moved into the suppl”.

We made changes as requested.

2. “The authors are not systematic on the “coloring” of the data. They consistently show deltaLepR as black but they jump between grey and white for f/f and LepR-cre. Not very important but there is no reason not to be systematic in this respect”.

We made changes as suggested.

3. “The authors show fat and lean masses as %, likely giving misleading effects of apparent loss of lean mass (fig S1B). The authors should show the data in g”.

We made changes as requested.

4. “Fig S1F is one of the most interesting outcomes of the study and it should definitely be upgraded to be included in the main figs”.

We made changes as requested.

5. “The word homeostasis in line 90 is perhaps not really optimal. The mice seem at least

concerning body temperature to be homeostatic, although at a lower body temp than normal mice”.

We deleted “homeostasis”, following the comments.

6. “Evidently, the same concerning showing the data per mouse and not per kg bw is relevant for fig 3 c (to be replaced by fig S2D). Similarly the fat content in fig. 3B and S2C should be in g not %”.

We made changes as requested.

7. “Line 124: energy expenditure is not regulated according to these data”

We deleted “energy expenditure”, following this comment.

8. “I wondered whether the stress that the authors suggest explains the lack of shift in RER also have masked any real differences in energy expenditure? The authors could mention this in the Discussion”.

We made changes as suggested. In the Discussion, we state that “We speculate that assay-related stress and/or other factors may influence energy expenditure and circadian rhythm, thereby masking difference between these two groups”.

9. “Same comments again for showing the total energy expenditure (not per kg bw) and the fat content in g in the main figures, for fig 4”.

We made changes as requested.

10. “Interesting that body temp is increased (fig 4 D). Again, these body temp data are underplayed by the authors”.

We added additional discussion about body temp, following this comment.

11. “Blood glucose in fig 3 and 4 insulin tolerance should be given as mg/ml, not %”.

We made changes as requested.

12. “Body temp is regulated by itself and is normally not secondary to BAT activity. E.g. when normal mice are transferred to cold (as in fig 5D) they mainly keep their body temp up by shivering. Thus, in my opinion, the change in body temperature setting observed here is a direct effect of the leptin system, not secondary to BAT activity. Thus, the “accordingly” line 155 is not correct”.

13. “Line 174: shouldn’t this be fig. 6E”?

We checked the results and confirmed that it is Fig. 6F. Fig. 6E shows baseline levels.

14. "(Comment omitted)"

15. "I am surprised concerning the UCP1 mRNA data in 5c (supposedly BAT) versus 1g (stated as iWAT). Normally when expressed versus house-keeping genes, UCP1 mRNA levels are 10-100-fold lower in iWAT than in BAT. Could the authors check their calculations"?

We checked qPCR Ct results and calculations. Ucp1 expression was dramatically lower in iWAT (Ct cycles: 27) relative to BAT (Ct cycle: 19).

16. "I was first surprised that the "controls" in 5D left and right were different – but then this may be due to the right ones being obese. This may be pointed out".

We added chow/HFD information to both the text and figure legends, following these comments.

17. "Of course the confusing sentences around line 250 are avoided when the unnormalized figs are shown instead".

We rewrote the discussion to clarify confusions.

18. "Line 277: do the authors mean "only slightly"?"

We replaced it with "only slightly".

19. "It is strange that the authors do not mention the very consistent and interesting data on body temperature regulation at all in the discussion. The study is in my opinion a very clear demonstration of the significance of leptin as a body temperature regulator. Perhaps this should also be included in the title".

We rewrote Discussion and expanded discussion about body temperature, following these comments. We also revised the title adding "adipose thermogenesis pathway".

20. "The authors several times write as if they have measured BAT thermogenesis, including in the abstract. This they have not done; they have "only" shown molecular evidence implying an attenuation of thermogenesis".

We revised descriptions and used thermogenic programs, which we directly assessed.

21. "Line 28 (abstract): this about energy expenditure is a little doubtful, considering what is discussed above".

We deleted "energy expenditure".

"With the above updates I feel that the paper will be a very interesting contribution".

We greatly appreciate Reviewer 2's constructive comments. Thanks!

Reviewers' Comments:

Reviewer #1:

Remarks to the Author:

The authors have provided data on bilateral injections of vectors and have sufficiently addressed other concerns raised in the previous round. No further comments from me.

Reviewer #2:

Remarks to the Author:

The authors have made clear improvements to the paper. I have now only a few points, mainly relating to my earlier points. I think the authors could well address these points in their final submission to the journal.

Referring to my earlier points:

Point 10: I am still surprised that the body temperature effects are not even mentioned in the Discussion section.

Point 15: I still think that the units for UCP1 mRNA levels should be harmonized between the figures. It is still not possible to understand what the units are, but e.g. the mean level of UCP1 in LepR-Cre in figure 5C should be set to 1.0 and the level in the right-hand part of that figure, as well as the data in suppl. Figure 1F, should be expressed accordingly (this means that the values in suppl 1F become very small as the level according to the comment by the authors is only about 1/250 of the level in BAT).

Point 19: It still seems that the authors do not distinguish between thermogenesis and body temperature control. It may be difficult to include body temperature control in the title but in the abstract (line 27) the authors still maintain that it is the lack of BAT that results in reduced regulated core body temperature. The word "resulting in" is thus misleading. Perhaps exchanging "resulting in" with "and lead to" or "and was associated with". Similar for line 77.

Response to Reviewers

Reviewer #1 (Remarks to the Author):

"The authors have provided data on bilateral injections of vectors and have sufficiently addressed other concerns raised in the previous round. No further comments from me".

We greatly appreciate reviewer 1's critiques.

Reviewer #2 (Remarks to the Author):

"The authors have made clear improvements to the paper. I have now only a few points, mainly relating to my earlier points. I think the authors could well address these points in their final submission to the journal".

"Referring to my earlier points":

"Point 10: I am still surprised that the body temperature effects are not even mentioned in the Discussion section".

We greatly appreciate reviewer 2's critiques. Following this comment, we expanded discussion about Sh2b1 regulation of body temperature. In the revision, we added "...These findings define LepR neuron Sh2b1 as a critical central regulator of thermogenesis and body temperature. Thus, we unveil an unrecognized leptin/LepR neuron Sh2b1/SNS/BAT/thermogenesis/body temperature axis. However, we cannot exclude the possibility that hypothermic Sh2b1 may increase thermogenesis and body temperature by an additional leptin-independent mechanism. For instance, Sh2b1 may enhance the ability of interleukin-6, a well-known pyrogenic cytokine, to increase thermogenesis and body temperature through enhancing the JAK2/Stat3 pathway. Furthermore, hypothalamic Sh2b1 may increase body temperature by a SNS-independent mechanism, perhaps by enhancing the ability of hypothalamic-pituitary-thyroid axis to increase thermogenesis and body temperature".

"Point 15: I still think that the units for UCP1 mRNA levels should be harmonized between the figures. It is still not possible to understand what the units are, but e.g. the mean level of UCP1 in LepR-Cre in figure 5C should be set to 1.0 and the level in the right-hand part of that figure, as well as the data in suppl. Figure 1F, should be expressed accordingly (this means that the values in suppl 1F become very small as the level according to the comment by the authors is only about 1/250 of the level in BAT)".

Following these comments, we replaced "Ucp1 mRNA" with "Relative Ucp1 mRNA levels (a.u.)" in Fig. 5c and Supple Fig. 1f, and stated in figure legends "a.u.: arbitrary units". We also added in the text that "Of note, absolute expression levels of Ucp1 was markedly higher in BAT than in WAT".

"Point 19: It still seems that the authors do not distinguish between thermogenesis and body temperature control. It may be difficult to include body temperature control in the title but in the

abstract (line 27) the authors still maintain that it is the lack of BAT that results in reduced regulated core body temperature. The word "resulting in" is thus misleading. Perhaps exchanging "resulting in" with "and lead to" or "and was associated with". Similar for line 77" .

We replaced "resulting in" with "leading to" as suggested.